# Do I Know This Entity? Knowledge Awareness and Hallucinations in Language Models

**Javier Ferrando**[1,2*]    **Oscar Obeso**[3*]    **Senthooran Rajamanoharan**    **Neel Nanda**

[1]U. Politècnica de Catalunya    [2]Barcelona Supercomputing Center    [3]ETH Zürich

## Abstract

Hallucinations in large language models are a widespread problem, yet the mechanisms behind whether models will hallucinate are poorly understood, limiting our ability to solve this problem. Using sparse autoencoders as an interpretability tool, we discover that a key part of these mechanisms is *entity recognition*, where the model detects if an entity is one it can recall facts about. Sparse autoencoders uncover meaningful directions in the representation space, these detect whether the model recognizes an entity, e.g. detecting it doesn't know about an athlete or a movie. This suggests that models might have self-knowledge: internal representations about their own capabilities. These directions are causally relevant: capable of steering the model to refuse to answer questions about known entities, or to hallucinate attributes of unknown entities when it would otherwise refuse. We demonstrate that despite the sparse autoencoders being trained on the base model, these directions have a causal effect on the chat model's refusal behavior, suggesting that chat finetuning has repurposed this existing mechanism. Furthermore, we provide an initial exploration into the mechanistic role of these directions in the model, finding that they disrupt the attention of downstream heads that typically move entity attributes to the final token.[1]

## 1 Introduction

Large Language Models (LLMs) have remarkable capabilities (Radford et al., 2019; Brown et al., 2020; Hoffmann et al., 2022; Chowdhery et al., 2023) yet have a propensity to hallucinate: generating text that is fluent but factually incorrect or unsupported by available information (Ji et al., 2023; Minaee et al., 2024). This significantly limits their application in real-world settings where factuality is crucial, such as healthcare. Despite the prevalence and importance of this issue, the mechanistic understanding of whether LLMs will hallucinate on a given prompt remains limited. While there has been much work interpreting factual recall (Geva et al., 2023; Nanda et al., 2023; Chughtai et al., 2024; Yu et al., 2023), it has mainly focused on the mechanism behind recalling known facts, not on hallucinations or refusals to answer, leaving a significant gap in our understanding.

Language models can produce hallucinations due to various factors, including flawed data sources or outdated factual knowledge (Huang et al., 2023). However, an important subset of hallucinations occurs when models are prompted to generate information they don't possess. We operationalize this phenomenon by considering queries about entities of different types (movies, cities, players, and songs). Given a question about an unknown entity, the model either hallucinates or refuses to answer. In this work, we find linear directions in the representation space that potentially encode a form of self-knowledge: assessing their own knowledge or lack thereof regarding specific entities. These directions are causally relevant for whether it refuses to answer. We note that the existence of this kind of knowledge awareness does not necessarily imply the existence of other forms of self-knowledge, and may be specific to the factual recall mechanism.

We find these directions using Sparse Autoencoders (SAEs) (Bricken et al., 2023; Huben et al., 2024). SAEs are an interpretability tool for finding a sparse, interpretable decomposition of model

---

*Equal contribution. Work done as part of the ML Alignment & Theory Scholars (MATS) Program. Correspondence to jferrandomonsonis@gmail.com, balcells.oscar@gmail.com.

[1]We make the codebase available at https://github.com/javiferran/sae_entities.

| Known Entity Latent Activations | Unknown Entity Latent Activations |
|---|---|
| Michael Jordan | Michael Joordan |
| When was the player LeBron James born? | When was the player Wilson Brown born? |
| He was born in the city of San Francisco | He was born in the city of Anthon |
| I just watched the movie 12 Angry Men | I just watched the movie 20 Angry Men |
| The Beatles song 'Yellow Submarine' | The Beatles song 'Turquoise Submarine' |

Table 1: Pair of sparse autoencoder latents that activate on known (left) and unknown entities (right) respectively. They fire consistently across entity types (movies, cities, songs, and players).

representations. They are motivated by the Linear Representation Hypothesis (Park et al., 2023; Mikolov et al., 2013): that interpretable properties of the input (features) such as sentiment (Tigges et al., 2023) or truthfulness (Li et al., 2023; Zou et al., 2023) are encoded as linear directions in the representation space, and that model representations are sparse linear combinations of these directions. We use Gemma Scope (Lieberum et al., 2024), which offers a suite of SAEs trained on every layer of Gemma 2 models (Team et al., 2024), and find internal representations that suggest to encode knowledge awareness in Gemma 2 2B and 9B. Additionally, we reproduce these findings for the Llama 3.1 8B model (Grattafiori et al., 2024) using LlamaScope's SAE suite (He et al., 2024), with results presented in Appendix Q.

Arditi et al. (2024) discovered that the decision to refuse a harmful request is mediated by a single direction. Building on this work, we demonstrate that a model's refusal to answer requests about attributes of entities (*knowledge refusal*) can similarly be steered with our found entity recognition directions. This finding is particularly intriguing given that Gemma Scope SAEs were trained on the base model on pre-training data. Yet, SAE-derived directions have a causal effect on knowledge-based refusal in the chat model-a behavior incentivized in the finetuning stage. This insight provides additional evidence for the hypothesis that finetuning often repurposes existing mechanisms (Jain et al., 2024; Prakash et al., 2024; Kissane et al., 2024).

Overall, our contributions are as follows:

- Using sparse autoencoders (SAEs) we **discover directions in the representation space on the final token of an entity, detecting whether the model can recall facts about the entity**, suggesting they encode a form of knowledge awareness.

- Our findings show that **entity recognition directions generalize across diverse entity types**: players, films, songs, cities, and more.

- We demonstrate that these directions **causally affect knowledge refusal in the chat model**, i.e. by steering with these directions, we can cause the model to hallucinate rather than refuse on unknown entities, and refuse to answer questions about known entities.

- We find that **unknown entity recognition directions disrupt the factual recall mechanism**, by suppressing the attention of attribute extraction heads, shown in prior work (Nanda et al., 2023; Geva et al., 2023) to be a key part of the mechanism.

- We go beyond merely understanding knowledge refusal, and find **SAE latents, seemingly representing uncertainty, that are predictive of incorrect answers**.

## 2 SPARSE AUTOENCODERS

Dictionary learning (Olshausen & Field, 1997) offers a powerful approach for disentangling features in superposition. Sparse Autoencoders (SAEs) have proven to be effective for this task (Sharkey et al., 2022; Bricken et al., 2023). SAEs project model representations $x \in \mathbb{R}^d$ into a larger dimensional space $a(x) \in \mathbb{R}^{d_{SAE}}$. In this work, we use the SAEs from Gemma Scope (Lieberum et al., 2024)[2], which use the JumpReLU SAE architecture (Rajamanoharan et al., 2024), which defines the

[2]We use the default sparsity for each layer, the ones available in Neuronpedia (Lin & Bloom, 2024).

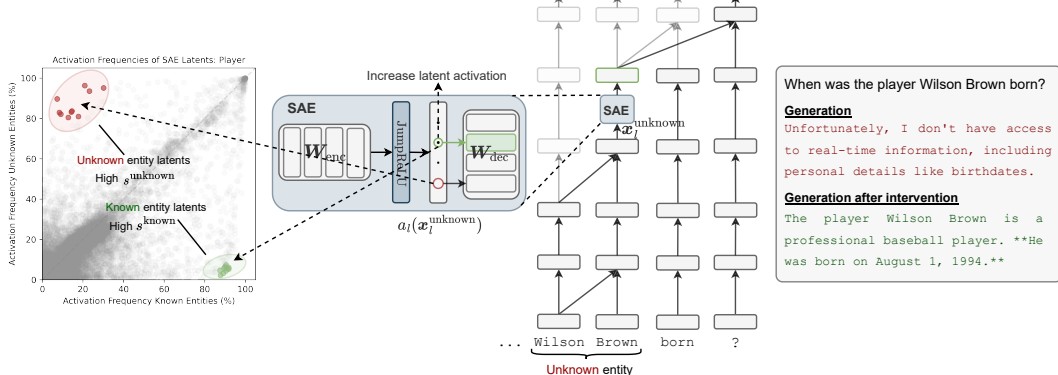

Figure 1: We identify SAE latents in the final token of the entity residual stream (i.e. hidden state) that almost exclusively activate on either unknown or known entities (scatter plot on the left). Modulating the activation values of these latents, e.g. increasing the known entity latent when asking a question about a made-up athlete increases the tendency to hallucinate.

function

$$\text{SAE}(\boldsymbol{x}) = a(\boldsymbol{x})\boldsymbol{W}_{\text{dec}} + \boldsymbol{b}_{\text{dec}}, \tag{1}$$

where

$$a(\boldsymbol{x}) = \text{JumpReLU}_\theta\big(\boldsymbol{x}\boldsymbol{W}_{\text{enc}} + \boldsymbol{b}_{\text{enc}}\big), \tag{2}$$

with the activation function (Erichson et al., 2019) $\text{JumpReLU}_\theta(\boldsymbol{x}) = \boldsymbol{x} \odot H(\boldsymbol{x} - \theta)$, composed by $H$, the Heaviside step function, and $\theta$, a learnable vector acting as a threshold. Intuitively, this is zero below the threshold, and then the identity, with a discontinuous jump at the threshold. $\boldsymbol{W}_{\text{enc}}$, $\boldsymbol{b}_{\text{enc}}$ and $\boldsymbol{W}_{\text{dec}}$, $\boldsymbol{b}_{\text{dec}}$ are the weight matrices and bias of the encoder and decoder respectively. We refer to *latent activation* to a component in $a(\boldsymbol{x})$, while we reserve the term *latent direction* to a (row) vector in the dictionary $\boldsymbol{W}_{\text{dec}}$.

Equation (1) shows that the model representation can be approximately reconstructed by a linear combination of the *SAE decoder latents*, which often represent monosemantic features (Huben et al., 2024; Bricken et al., 2023; Templeton et al., 2024; Gao et al., 2024). By incorporating a sparsity penalty into the training loss function, we can constrain this reconstruction to be a sparse linear combination, thereby enhancing interpretability:

$$\mathcal{L}(\boldsymbol{x}) = \underbrace{\|\boldsymbol{x} - \text{SAE}(\boldsymbol{x})\|_2^2}_{\mathcal{L}_{\text{reconstruction}}} + \underbrace{\lambda \|a(\boldsymbol{x})\|_0}_{\mathcal{L}_{\text{sparsity}}}. \tag{3}$$

**Steering with SAE Latents.** Recall from Equation (1) that SAEs reconstruct a model's representation as $\boldsymbol{x} \approx a(\boldsymbol{x})\boldsymbol{W}_{\text{dec}} + \boldsymbol{b}_{\text{dec}}$. This means that the reconstruction is a linear combination of the decoder latents (rows) of $\boldsymbol{W}_{\text{dec}}$ plus a bias, i.e. $\boldsymbol{x} \approx \sum_j a_j(\boldsymbol{x})\boldsymbol{W}_{\text{dec}}[j, :]$. Thus, increasing/decreasing the activation value of an SAE latent, $a_j(\boldsymbol{x})$, is equivalent to doing activation steering (Turner et al., 2023) with the decoder latent vector, i.e. updating the residual stream as follows:

$$\boldsymbol{x}^{\text{new}} \leftarrow \boldsymbol{x} + \alpha \boldsymbol{d}_j. \tag{4}$$

## 3 METHODOLOGY

To study how language models reflect knowledge awareness about entities, we build a dataset with four different entity types: (basketball) players, movies, cities, and songs from Wikidata (Vrandečić & Krötzsch, 2024). For each entity, we extract associated attributes available in Wikidata. Then, we create templates of the form (entity type, entity name, relation, attribute) and prompt Gemma 2 2B and 9B models (Team et al., 2024) to predict the attribute given (entity type, relation, entity name), for instance:

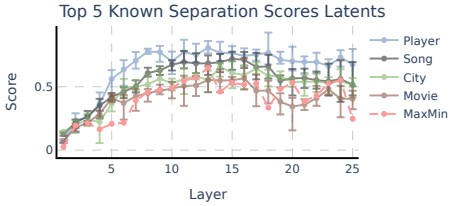 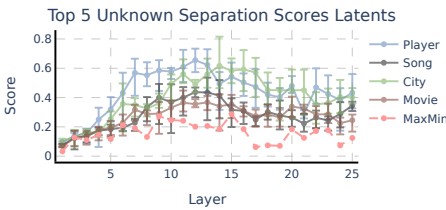

Figure 2: Layerwise evolution of the Top 5 latents in Gemma 2 2B SAEs, as measured by their known (left) and unknown (right) latent separation scores ($s^{\text{known}}$ and $s^{\text{unknown}}$). Error bars show maximum and minimum scores. MaxMin (red line) refers to the minimum separation score across entities of the best latent. This represents how entity-agnostic is the most general latent per layer. In both cases, the middle layers provide the best-performing latents.

$$\text{Entity type} \downarrow \qquad\qquad \downarrow \text{Relation}$$
$$\text{The } \texttt{movie 12 Angry Men was directed by } \_\_ \qquad (5)$$
$$\text{Entity name} \uparrow \qquad\qquad\qquad \uparrow \text{Attribute}$$

We then categorize entities into 'known' or 'unknown'. Known entities are those where the model gets at least two attributes correct, while unknown are where it gets them all wrong, we discard any in-between. To measure correctness we use fuzzy string matching.[3] See Appendix A for a description of the process. We acknowledge that this methodology might introduce some labeling inaccuracies, as the model could 'guess' some attributes despite not knowing about the entity or fail to recall the specific attributes we consider while knowing about the entity. However, our primary objective is to achieve a reasonable differentiation between entities rather than striving for perfect classification accuracy. Finally, we split the entities into train/validation/test (50%, 10%, 40%) sets.

We run the model on the set of prompts containing known and unknown entities. Inspired by Meng et al. (2022a); Geva et al. (2023); Nanda et al. (2023) we use the residual stream of the final token of the entity, $\boldsymbol{x}^{\text{known}}$ and $\boldsymbol{x}^{\text{unknown}}$. In each layer ($l$), we compute the activations of each latent in the SAE, i.e. $a_{l,j}(\boldsymbol{x}_l^{\text{known}})$ and $a_{l,j}(\boldsymbol{x}_l^{\text{unknown}})$. For each latent, we obtain the fraction of the time that it is active (i.e. has a value greater than zero) on known and unknown entities respectively:

$$f_{l,j}^{\text{known}} = \frac{\sum_i^{N^{\text{known}}} \mathbb{1}[a_{l,j}(\boldsymbol{x}_{l,i}^{\text{known}}) > 0]}{N^{\text{known}}}, \quad f_{l,j}^{\text{unknown}} = \frac{\sum_i^{N^{\text{unknown}}} \mathbb{1}[a_{l,j}(\boldsymbol{x}_{l,i}^{\text{unknown}}) > 0]}{N^{\text{unknown}}}, \quad (6)$$

where $N^{\text{known}}$ and $N^{\text{unknown}}$ are the total number of prompts in each subset. Then, we take the difference, obtaining the *latent separation scores* $s_{l,j}^{\text{known}} = f_{l,j}^{\text{known}} - f_{l,j}^{\text{unknown}}$ and $s_{l,j}^{\text{unknown}} = f_{l,j}^{\text{unknown}} - f_{l,j}^{\text{known}}$, for detecting known and unknown entities respectively.

## 4 SPARSE AUTOENCODERS UNCOVER ENTITY RECOGNITION DIRECTIONS

We find that the separation scores of some of the SAE latents in the training set are high, i.e. they fire almost exclusively on tokens of either known or unknown entities, as depicted in the scatter plot in Figure 1 for Gemma 2 2B and Figure 8, Appendix C for Gemma 2 9B. An interesting observation is that latent separation scores reveal a consistent pattern across all entity types, with scores increasing throughout the model and reaching a peak around layer 9 before plateauing (Figure 2). This indicates that *latents better distinguishing between known and unknown entities are found in the middle layers*.

We also examine the level of generality of the latents by measuring their minimum separation score across entity types ($t$): players, song, cities and movies. A high minimum separation score indicates that a latent performs robustly across entity types, suggesting strong generalization capabilities. For this purpose, for each layer ($l$) we compute $\text{MaxMin}^{\text{known},l} = \max_j \min_t s_{l,j}^{\text{known},t}$, and similarly for unknown entities. The increasing trend shown in the MaxMin (red) line in Figure 2 for Gemma 2 2B and in Figure 9, Appendix D for Gemma 2 9B suggests that more *generalized* latents—those that distinguish between known and unknown entities across various entity types—are concentrated

---
[3]https://github.com/seatgeek/thefuzz.

in these intermediate layers. This observation is replicated on Llama 3.1 8B in Appendix Q. This finding points to a hierarchical organization of entity representation within the model, with more specialized, worse quality, latents in earlier layers and more generalized, higher quality entity-type-agnostic features emerging in the middle layers.

Next, we compute the minimum separation scores by considering every SAE latent in every layer, i.e. $\min_t s_{l,j}^{\text{known},t}$ for $1 \leq l \leq L$ and $1 \leq j \leq d_{SAE}$, and equivalently for unknown entities. To ensure specificity to entity tokens, we exclude latents that activate frequently (>2%) on random tokens sampled from the Pile dataset (Gao et al., 2020). The latents with highest minimum separation scores exhibit the most generalized behavior out of all latents, and will be the focus of our subsequent analysis:

$$\text{known entity latent} = \arg\max_{l,j} \underbrace{\min_t s_{l,j}^{\text{known},t}}_{\substack{\text{min known separation score} \\ \text{of latent } l, j \text{ across entity types}}} \quad \text{and} \quad \text{unknown entity latent} = \arg\max_{l,j} \underbrace{\min_t s_{l,j}^{\text{unknown},t}}_{\substack{\text{min unknown separation score} \\ \text{of latent } l, j \text{ across entity types}}} . \quad (7)$$

Table 1 demonstrates the activation patterns of the Gemma 2 2B topmost known entity latent on prompts with well-known entities (left), and the patterns for the topmost unknown entity latent (right), firing across entities of different types that cannot be recognized. In Appendix B we provide the activations of these latents on sentences containing a diverse set of entity types, suggesting that indeed they are highly general. To validate these latents' reliability, we analyze their activation frequencies on 283 song titles released after the models' knowledge cutoff (August 2024). As hypothesized, unknown entity latents show higher activation rates, while known entity latents exhibit lower activation frequencies (Appendix R). While we acknowledge potential overlap between these song titles and pre-training data, the consistent activation patterns across multiple models strengthen our confidence in these latents' ability to distinguish between known and unknown information. In the following sections, we explore how these primary entity recognition latents influence the model's overall behavior.

## 5 ENTITY RECOGNITION DIRECTIONS CAUSALLY AFFECT KNOWLEDGE REFUSAL

We define *knowledge refusal* as the model declining to answer a question due to reasons like a lack of information or database access as justification, rather than safety concerns. To quantify knowledge refusals, we adapt the factual recall prompts as in Example 5 into questions:

$$\overset{\text{Relation} \downarrow}{\underset{\text{Entity name} \uparrow}{\text{Who directed the movie 12 Angry Men}}} \overset{\text{Entity type} \downarrow}{\underset{\text{Attribute} \uparrow}{\text{? \_\_\_}}} \quad (8)$$

and we define a set of common knowledge refusal completions and detect if any of these occur with string matching, e.g. *'Unfortunately, I don't have access to real-time information...'.* Gemma 2 includes both a base model, and a fine-tuned chat (i.e. instruction tuned) model. In Section 4 we found the entity recognition latents by studying the base model, but here focus on the chat model, as they have been explicitly fine-tuned to perform knowledge refusal where appropriate (Team et al., 2024)[4], and the factuality of chat models is highly desirable.

We hypothesize that entity recognition directions could be used by chat models to induce knowledge refusal. To evaluate this, we use a test set sample of 100 questions about unknown entities, and measure the number of times the model refuses by steering (as in Equation (4)) with the entity recognition latents the last token of the entity and the following end-of-instruction-tokens.[5] Figure 3 (left)

---

[4]The Gemma 2 technical report (Team et al., 2024) mentions *"including subsets of data that encourage refusals to minimize hallucinations improves performance on factuality metrics"*. This pattern is consistent with recent language models, such as Llama 3.1 (Dubey et al., 2024), where the explicit finetuning process for knowledge refusal has been documented.

[5]We use a validation set to select an appropriate steering coefficient $\alpha$. In Appendix G we show generations of Gemma 2B IT with different steering coefficients. We select $\alpha \in [400, 550]$, which corresponds to around two times the norm of the residual stream in the layers where the entity recognition latents are present (Appendix E).

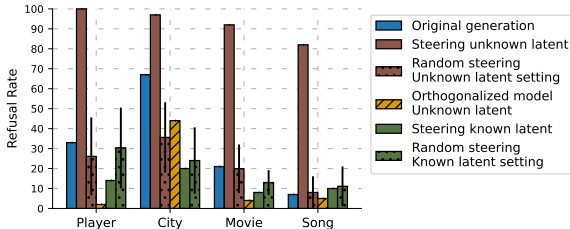 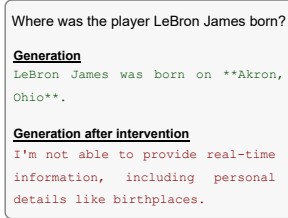

Figure 3: **Left**: Number of times Gemma 2 2B refuses to answer in 100 queries about unknown entities. We examine the unmodified original model, the model steered with the known entity latent and unknown entity latent, and the model with the unknown entity latent projected out of its weights (referred to as Orthogonalized model). The mean and standard deviation of steering with 10 random latents are shown for comparison. **Right**: This example illustrates the effect of steering with the unknown entity recognition latent (same as in Table 1). The steering induces the model to refuse to answer about a well-known basketball player.

illustrates the original model refusal rate (blue bar), showing some refusal across entity types. We see that the entity recognition SAE latents found in the base model transfer to the chat model, and increasing the unknown entity latent induces almost 100% refusal across all entity types in Gemma 2 2B. Conversely, increasing the known entity latent activation slightly reduces refusal rates. We also include an *Orthogonalized model* baseline, which consists of doing weight orthogonalization (Arditi et al., 2024) on every matrix writing to the residual stream. Weight orthogonalization modifies each row of a weight matrix to make it perpendicular to a specified direction vector $d$. This is achieved by subtracting the component of each row that is parallel to $d$:

$$W_{\text{out}}^{\text{new}} \leftarrow W_{\text{out}} - W_{\text{out}} d^\intercal d. \tag{9}$$

By doing this operation on every output matrix in the model we ensure no component is able to write into that direction. The resulting orthogonalized model with the top unknown entity direction exhibits a large reduction in refusal responses, suggesting this direction plays a crucial role in the model's knowledge refusal behavior. We also include the average refusal rate after steering with 10 differents random latents, using the same configuration (layer and steering coefficient) that the known and unknown entity latents respectively. Additional analysis on the Gemma 2 9B model and Llama 3.1 8B are detailed in Sections F and Q revealing similar patterns, albeit with less pronounced effects compared to the 2B model.

Figure 3 (right) shows a refusal response for a well-known basketball player generated by steering with the unknown entity latent. In Figure 1 (right) we observe that when asked about a non-existent player, Wilson Brown, the model without intervention refuses to answer. However, steering with the known entity latent induces a hallucination.

## 6   MECHANISTIC ANALYSIS

**Entity Recognition Directions Regulate Attention to Entity.** In the previous section, we saw that entity recognition latents had a causal effect on knowledge refusal. Here, we look at how they affect the factual recall mechanism (*aka* circuit) in prompts of the format of Example 5. This has been well studied before on other language models (Nanda et al., 2023; Geva et al., 2023; Meng et al., 2022a). We replicate the approach of Nanda et al. (2023) on Gemma 2 2B and 9B and find a similar circuit. Namely, early attention heads merge the entity's name into the last token of the entity, and downstream attention heads extract relevant attributes from the entity and move them to the final token position (Figure 4 (a, b)), this pattern holds across various entity types and model sizes (Appendix I and Appendix J). To do the analysis, we perform activation patching (Geiger et al., 2020; Vig et al., 2020; Meng et al., 2022a) on the residual streams and attention heads' outputs (see Appendix H for a detailed explanation on the method). We use the denoising setup (Heimersheim &

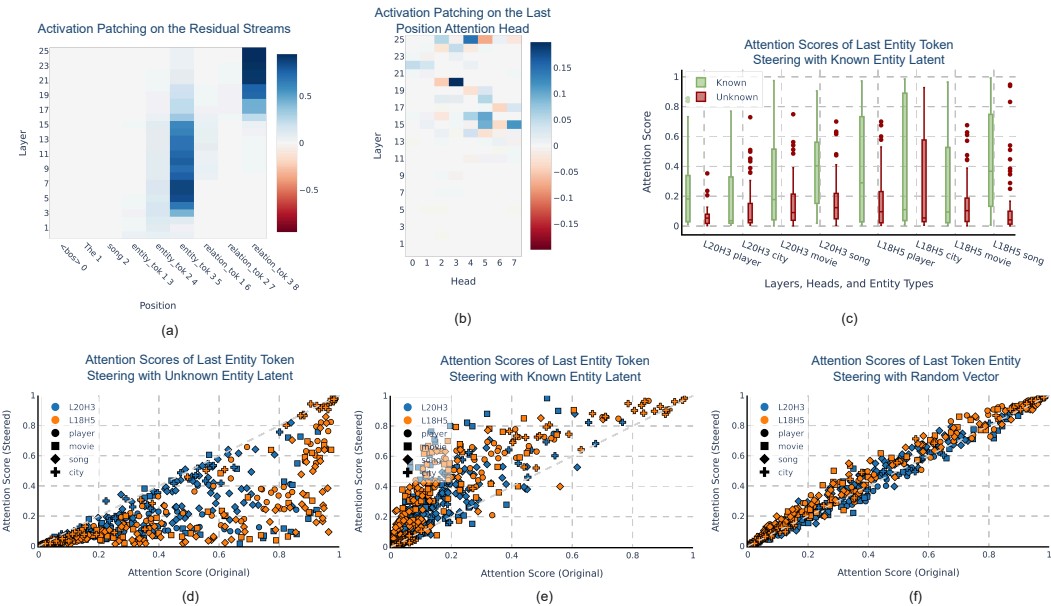

Figure 4: (a,b) Activation patching on the residual streams and the output of attention heads in the last position (song entities). We patch clean (from known entities prompts) representations into a corrupted forward pass (from unknown entities prompts) and measure the logit difference recovered. (c) Attention paid from the last position to the last token of the entity is greater when faced with a known entity in attribute-extraction heads. (d,e,f) Effect on attention scores, as in (c), after steering the last token of the entity with the unknown entity latent (d), known entity latent (e), and a random vector with same norm (f).

Nanda, 2024), where we patch representations from a clean run (with a known entity) and apply it over the run with a corrupted input (with an unknown entity).[6]

Expanding on the findings of Yuksekgonul et al. (2024), who established a link between prediction accuracy and attention to the entity tokens, our study reveals a large disparity in attention between known and unknown entities, for instance the attribute extraction heads L18H5 and L20H3 (Figure 4 (c)), which are overall relevant across entity types in Gemma 2 2B (see example of attributes extracted by these heads in Appendix L). Notably, attention scores are higher when faced with a known entity. This suggests that the detected entity recognition latents might influence the attention mechanism through the 'keys' computation to induce this behavior. To evaluate this hypothesis we steer the residual stream with the found latents on the last token of the entity, and measure the attention scores of the entity tokens. We observe a causal relationship between the entity recognition latents and the attention patterns of the attention heads downstream, being more pronounced in the attribute extraction heads. Steering with the top unknown entity latent reduces the attention to entity, even in prompts with a known entity (Figure 4 (d)), while steering with the known entity latent increases the attention scores (Figure 4 (e)). We show in Figure 4 (f) the results of steering with a random unit vector baseline for comparison, and in Appendix K when steering with a random SAE latent. In Appendix M we illustrate the average attention score change to the entity tokens after steering on the residual streams of the last token of the entities in Gemma 2 2B, 9B, and Llama 3.1 8B with the top 3 known and unknown entity latents. The results reveal an increase/decrease attention score across upper layer heads, with the 9B model showing more subtle effects when steered using unknown latents.

These results provide compelling evidence that the entity recognition SAE latent directions play a crucial role in regulating the model's attention mechanisms, and thereby their ability to extract attributes.

---

[6]We show the proportion of logit difference recovered after each patch in Figure 4 (a). A recovered logit difference of 1 indicates that the prediction after patching is the same as the original prediction in the clean run.

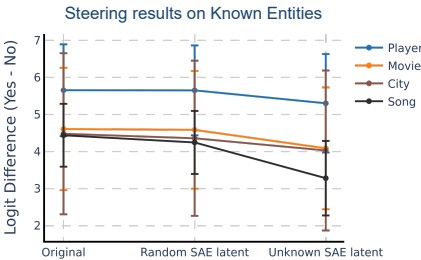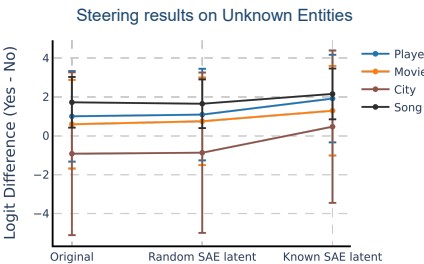

Figure 5: Logit difference between "Yes" and "No" predictions on the question "Are you sure you know the {entity_type} {entity_name}? Answer yes or no." after steering with unknown (left) and known (right) entity recognition latents.

**Early Entity Recognition Directions Regulate Expressing Knowledge Uncertainty.** We have shown that the entity recognition latents causally affect the model's knowledge refusal, implicitly using its knowledge of whether it recognises an entity, but not whether they are used when explicitly asking a model whether it recognises an entity. To investigate this, we use the following prompt structure:

```
Are you sure you know the {entity_type} {entity}? Answer yes or no. Answer: ___
```
(10)

We then steer the residual streams of the last token of the entity by upweighting the entity recognition latents. In Figure 5 we show the results on the logit difference between Yes and No responses. The left plot illustrates the effect of steering known entities prompts with the unknown entity latent. This intervention results in a reduction of the logit difference. For comparison, we include a random baseline where we apply a randomly sampled SAE latent with the same coefficient. In the right plot, we steer unknown entities prompts with the known entity latent. Despite the model's inherent bias towards Yes predictions for unknown entities (indicated by positive logit differences in the 'Original' column), which indicates the model struggles to accurately express their uncertainty (Yona et al., 2024), this intervention leads to a positive shift in the logit difference, suggesting that the entity recognition latents, although slightly, have an effect on the expression of uncertainty about knowledge of entities. A similar pattern can be observed in Gemma 2 9B (Appendix N).

## 7 UNCERTAINTY DIRECTIONS

Having studied how base models represent features for entity recognition, we now explore internal representations that may differentiate between correct and wrong answers. Our investigation focuses on chat models, which are capable of refusing to answer, and we search for directions in the representation space signaling uncertainty or lack of knowledge potentially indicative of upcoming errors. For this analysis we use our entities dataset, and exclude instances where the model refuses to respond, and leave only prompts that elicit either correct predictions or errors from the model.

Our study focuses on the study of the residual streams *before* the answer. We hypothesize that end-of-instruction tokens, which always succeed the instruction, may aggregate information about the whole question (Marks & Tegmark, 2023).[7] We select the token model and use examples such as:

```
<start_of_turn>user\nWhen was the player Wilson Brown born?<end_of_turn>\n<start_of_turn>model\n
```
(11)

For each entity type and layer with available SAE we extract the representations of the model residual stream, for both correct and mistaken answers, and gather the SAE latent activations. We are interested in seeing whether there are SAE latents that convey information about how unsure or uncertain the model is to answer to a question, but still fails to refuse, giving rise to hallucinations. We divide the dataset of prompts into train/validation/test sets (50%, 10%, 40%).

To capture subtle variations in model uncertainty, which may be represented even when attributes are correctly recalled, we focus on quantifying differences in activation levels between correct and

---

[7]This concept was termed by Tigges et al. (2023) as the 'summarization motif'.

| **'Unknown' Latent Activations** |
|---|
| "Apparently one or two people were shooting or shooting at each other for reasons unknown when eight people were struck by the gunfire |
| ...and the Red Cross all responded to the fire. The cause of the fire remains under investigation. |
| The Witcher Card Game will have another round of beta tests this spring (platforms TBA) |
| His condition was not disclosed, but police said he was described as stable. |

Table 2: Activations of the Gemma 2B IT 'unknown' latent on the maximally activating examples provided by Neuropedia (Lin & Bloom, 2024).

incorrect responses. For each latent, we compute the t-statistic in the training set using two activation samples: $a_{l,j}(\boldsymbol{x}_l^{\text{correct}})$ for correct responses and $a_{l,j}(\boldsymbol{x}_l^{\text{error}})$ for incorrect ones. The t-statistic measures how different the two sample means are from each other, taking into account the variability within the samples:

$$\text{t-statistic}_{l,j} = \frac{\mu(a_{l,j}(\boldsymbol{x}_l^{\text{correct}})) - \mu(a_{l,j}(\boldsymbol{x}_l^{\text{error}}))}{\sqrt{\frac{\sigma(a_{l,j}(\boldsymbol{x}_l^{\text{correct}}))^2}{n^{\text{correct}}} + \frac{\sigma(a_{l,j}(\boldsymbol{x}_l^{\text{error}}))^2}{n^{\text{error}}}}}. \tag{12}$$

We use a pre-trained SAE for the 13th layer (out of 18) of Gemma 2B IT[8], and the available Gemma Scope SAEs for Gemma 2 9B IT, at layers 10, 21, and 32 (out of 42). Our approach for detecting top latents, similar to the entity recognition method described in Section 4 focuses on the top latents with the highest minimum t-statistic score across entities, representing the most general latents. We split the dataset into train and test sets, and use the training set to select the top latents. The left panel of Figure 6 reveals a distinct separation between the latent activations at the model token when comparing correct versus incorrect responses in the test set. Using this latent as a classifier, it achieves 73.2 AUROC score, and by calibrating the decision threshold on a validation set, it gets an F1 score of 72. See Appendix P with separated errors by entity type. Table 2 illustrates the activations of the highest-scoring latent in Gemma 2B IT's SAE on a large text corpus (Penedo et al., 2024)[9], showing it triggers on text related to uncertainty or undisclosed information. Figure 6 (right) illustrates the top tokens with higher logit increase by this latent, further confirming its association with concepts of unknownness.[10] Similar latent separations between correct and incorrect answers can also be observed in Gemma 2 9B IT (Appendix O).

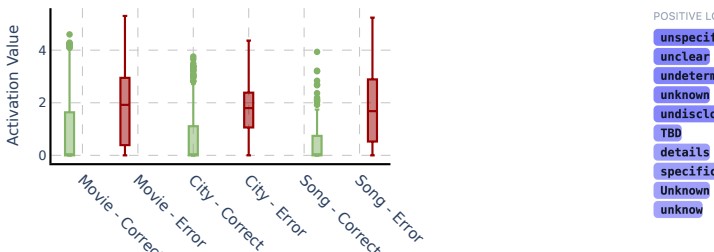

Figure 6: **Left**: Activation values of the Gemma 2B IT 'unknown' latent on correct and incorrect responses. **Right**: Top 10 tokens with the highest logit increase by the 'unknown' latent influence.

# 8 RELATED WORK

Recent advances in mechanistic interpretability in language models (Ferrando et al., 2024) have shed light on the factual recall process in these systems. Key discoveries include the aggregation of entity tokens (Nanda et al., 2023), the importance of early MLPs for entity processing (Meng et al.,

---

[8] https://huggingface.co/jbloom/Gemma-2b-IT-Residual-Stream-SAEs. We note that Gemma Scope doesn't provide SAEs for Gemma 2 2B IT.

[9] https://huggingface.co/datasets/HuggingFaceFW/fineweb.

[10] We omit the players category since Gemma 2B IT refuses to almost all of those queries.

2022b), and the identification of specialized extraction relation heads (Geva et al., 2023; Chughtai et al., 2024). Despite these insights, there remains a significant gap in our understanding of the mechanisms underlying failures in attribute extraction leading to hallucinations. Gottesman & Geva (2024) demonstrated that the performance of probes trained on the residual streams of entities correlates with the model's ability to answer questions about them accurately. Yuksekgonul et al. (2024) established a link between increased attention to entity tokens and improved factual accuracy. (Yu et al., 2024) proposed two mechanisms for non-factual hallucinations: inadequate entity enrichment in early MLPs and failure to extract correct attributes in upper layers. Our research aligns with studies on hallucination prediction (Kossen et al., 2024; Varshney et al., 2023), particularly those engaging with model internals (CH-Wang et al., 2024; Azaria & Mitchell, 2023). Previous work has trained probes to predict truthfulness of the produced outputs (Li et al., 2023) with Joshi et al. (2024) showing this can be detected in activation space before the model generation, which can be related to our results on 'uncertainty directions' discovered in Section 7. Additionally, our work contributes to the growing body of literature on practical applications of sparse autoencoders, as investigated by Marks et al. (2024); Krzyzanowski et al. (2024). While the practical applications of sparse autoencoders in language model interpretation are still in their early stages, our research demonstrates their potential.

## 9 CONCLUSIONS

In this paper, we use sparse autoencoders to identify directions in the model's representation space that suggest the presence of encoded knowledge awareness about entities. These directions, found in the base model, are causally relevant to the knowledge refusal behavior in the chat-based model. We demonstrated that, by manipulating these directions, we can control the model's tendency to refuse answers or hallucinate information. We also provide insights into how the entity recognition directions influence the model behavior, such as regulating the attention paid to entity tokens, and their influence in expressing knowledge uncertainty. Finally, we uncover directions representing model uncertainty to specific queries, capable of discriminating between correct and mistaken answers. While our primary focus in this work centers on the representation of knowledge awareness and uncertainty, the methodology we present for discovering these latent directions is generalizable to any other type of binary (Section 3) and continuous (Section 7) features. This work contributes to our understanding of language model behavior and opens avenues for improving model reliability and mitigating hallucinations.

### ACKNOWLEDGMENTS

This work was conducted as part of the ML Alignment & Theory Scholars (MATS) Program. We want to express our sincere gratitude to McKenna Fitzgerald for her guidance and support during the program, to Matthew Wearden for his thoughtful feedback on the manuscript, and to Wes Gurnee for initial discussions that helped shape this work. We want to extend our gratitude to Adam Karvonen, Can Rager, Bart Bussmann, Patrick Leask and Stepan Shabalin for the valuable input during MATS. Lastly, we thank the entire MATS and Lighthaven staff for creating the environment that made this research possible. Portions of this work were supported by the Long Term Future Fund. Javier Ferrando is supported by the fellowship within the "Generación D" initiative, Red.es, Ministerio para la Transformación Digital y de la Función Pública, for talent atraction (C005/24-ED CV1). Funded by the European Union NextGenerationEU funds, through PRTR. Lastly, we appreciate the anonymous reviewers for their useful comments.

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

## A  ENTITY DIVISION INTO KNOWN AND UNKNOWN

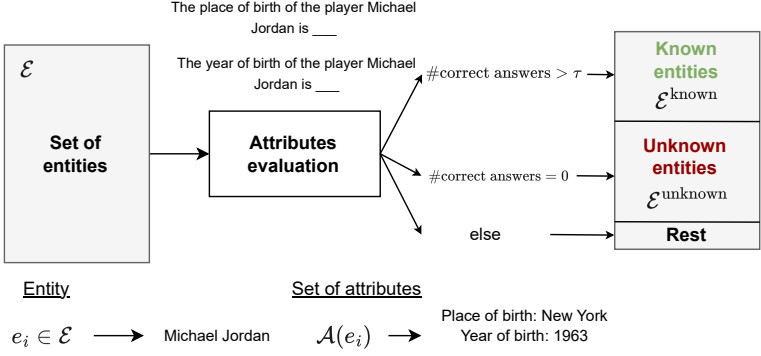

Figure 7: Pipeline for classifying entities as known or unknown. Each entity $e_i \in \mathcal{E}$ is evaluated by querying the language model about a set of attributes $\mathcal{A}(e_i)$. Classification as known or unknown is based on the accuracy of the model's responses. In this work we set the threshold $\tau = 1$.

| Entity Type | Number of entities | Attributes |
|---|---|---|
| Player | 7487 | Birthplace, birthdate, teams played |
| Movie | 10895 | Director, screenwriter, release date, genre, duration, cast |
| City | 7904 | Country, population, elevation, coordinates |
| Song | 8448 | Artist, album, publication year, genre |

Table 3: Entity types and attributes extracted from Wikidata.

# B  ENTITY RECOGNITION LATENTS ON DIVERSE ENTITIES

| Known Entity Latent Activations | Unknown Entity Latent Activations |
|---|---|
| Many people use Twitter to share their thoughts. | Many people use Twitter to share their thoughts. |
| L'Oréal is a large cosmetics and beauty company. | L'Oréal is a large cosmetics and beauty company. |
| The Mona Lisa is displayed in the Louvre museum. | The Mona Lisa is displayed in the Louvre museum. |
| Many people use Snapchat for sharing photos and short videos. | Many people use Snapchat for sharing photos and short videos. |
| The Acropolis is an ancient citadel in Athens. | The Acropolis is an ancient citadel in Athens. |
| The Galapagos Islands are known for their unique wildlife. | The Galapagos Islands are known for their unique wildlife. |
| Many people use Dropbox for cloud storage. | Many people use Dropbox for cloud storage. |
| The pyramids of Giza were built by ancient Egyptians. | The pyramids of Giza were built by ancient Egyptians. |
| Walmart is the world's largest company by revenue. | Walmart is the world's largest company by revenue. |
| FedEx is a multinational delivery services company. | FedEx is a multinational delivery services company. |
| Many people use Instagram to share photos. | Many people use Instagram to share photos. |
| The Neuschwanstein Castle inspired Disney's Sleeping Beauty Castle. | The Neuschwanstein Castle inspired Disney's Sleeping Beauty Castle. |
| The theory of gravity was developed by Isaac Newton. | The theory of gravity was developed by Isaac Newton. |
| Sony is known for its electronics and entertainment products. | Sony is known for its electronics and entertainment products. |
| Many people use Skype for voice and video calls. | Many people use Skype for voice and video calls. |
| The Sistine Chapel is famous for its frescoes by Michelangelo. | The Sistine Chapel is famous for its frescoes by Michelangelo. |
| The Andes are the longest continental mountain range in the world. | The Andes are the longest continental mountain range in the world. |
| The theory of evolution was proposed by Charles Darwin. | The theory of evolution was proposed by Charles Darwin. |
| Many people use Shopify for e-commerce platforms. | Many people use Shopify for e-commerce platforms. |
| Honda is known for its motorcycles and automobiles. | Honda is known for its motorcycles and automobiles. |

Table 4: Activations of Gemma 2 2B entity recognition latents on LLM generated data.

| Known Entity Latent Activations | Unknown Entity Latent Activations |
|---|---|
| Druids commune with nature in the sacred grove of Elthalas. | Druids commune with nature in the sacred grove of Elthalas. |
| Adventurers seek the lost treasure of King Zephyrion. | Adventurers seek the lost treasure of King Zephyrion. |
| The Thaumaturge's Guild specializes in Aether manipulation. | The Thaumaturge's Guild specializes in Aether manipulation. |
| The Vorpal Blade was forged by the legendary Jabberwock. | The Vorpal Blade was forged by the legendary Jabberwock. |
| The Hivemind of Xarzith threatens galactic peace. | The Hivemind of Xarzith threatens galactic peace. |
| Travelers must appease the Stormcaller to cross the Tempest Sea. | Travelers must appease the Stormcaller to cross the Tempest Sea. |
| Archaeologists unearthed artifacts from the Zanthar civilization. | Archaeologists unearthed artifacts from the Zanthar civilization. |
| Sailors fear the treacherous waters of the Myroskian Sea. | Sailors fear the treacherous waters of the Myrosian Sea. |
| Scientists studied the unique properties of Quixium alloy. | Scientists studied the unique properties of Quixium alloy. |
| The Glibberthorn plant is known for its healing properties. | The Glibberthorn plant is known for its healing properties. |
| The Voidwalker emerged from the Abyssal Rift. | The Voidwalker emerged from the Abyssal Rift. |
| Alchemists seek to create the legendary Philosopher's Stone. | Alchemists seek to create the legendary Philosopher's Stone. |
| Pilgrims seek enlightenment at the Temple of Ethereal Wisdom. | Pilgrims seek enlightenment at the Temple of Ethereal Wisdom. |
| Pilots navigate through the treacherous Astral Maelstrom. | Pilots navigate through the treacherous Astral Maelstrom. |
| Merchants trade rare gems in the bazaars of Khalindor. | Merchants trade rare gems in the bazaars of Khalindor. |
| Scholars study ancient texts at the University of Arcanum. | Scholars study ancient texts at the University of Arcanum. |
| The Vexnor device revolutionized quantum computing. | The Vexnor device revolutionized quantum computing. |
| The Whispering Woods are guarded by the Sylvani. | The Whispering Woods are guarded by the Sylvani. |
| The Ethereal Conclave governs the realm of spirits. | The Ethereal Conclave governs the realm of spirits. |
| The Quantum Forge harnesses the power of Nullstone. | The Quantum Forge harnesses the power of Nullstone. |

Table 5: Activations of Gemma 2 2B entity recognition latents on LLM generated data.

## C GEMMA 2 9B LATENTS ACTIVATION FREQUENCIES ON KNOWN AND UNKNOWN PROMPTS

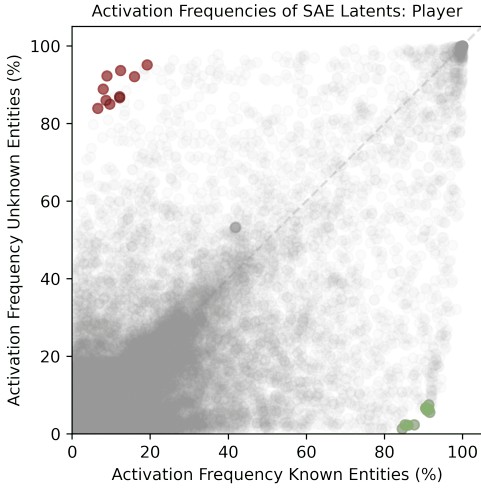

Figure 8: Activation frequencies of Gemma 2 9B SAE latents on known and unknown Prompts, in player entity type.

## D GEMMA 2 9B LAYERWISE EVOLUTION OF THE TOP 5 LATENTS

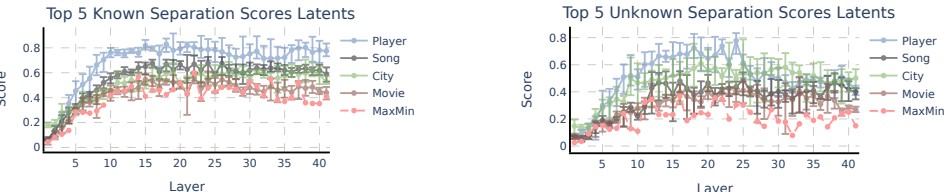

Figure 9: Gemma 2 9B layerwise evolution of the Top 5 latents, as measured by their known (left) and unknown (right) latent separation scores ($s^{\text{known}}$ and $s^{\text{unknown}}$). Error bars show maximum and minimum scores. MaxMin (red line) refers to the minimum separation score across entities of the best latent. This represents how entity-agnostic is the most general latent per layer. In both cases, middle layers provide the best-performing latents.

# E  NORM RESIDUAL STREAMS

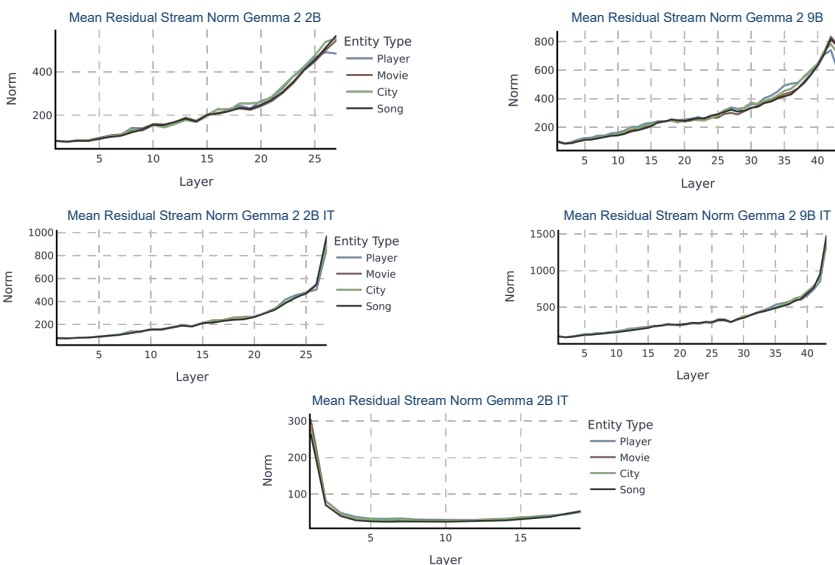

Figure 10: Norm of the residual streams of the last token of the entity across layers of the different Gemma models.

# F  REFUSAL RATES GEMMA 2 9B

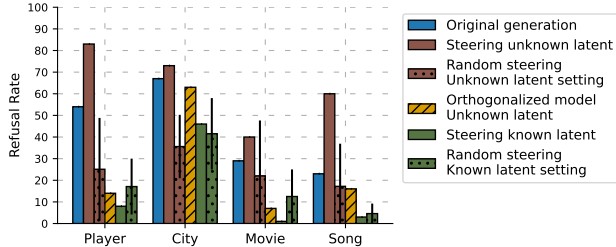

Figure 11: **Left**: Number of times Gemma 2 9B refuses to answer in 100 queries about unknown entities. We examine the unmodified original model, the model steered with the known entity latent and unknown entity latent, and the model with the unknown entity latent projected out of its weights (referred to as Orthogonalized model). The mean and standard deviation of steering with 10 random latents are shown for comparison. **Right**: This example illustrates the effect of steering with the unknown entity recognition latent. The steering induces the model to refuse to answer about a well-known basketball player.

## G    EXAMPLE OF GENERATIONS STEERING WITH DIFFERENT COEFFICIENTS

| **Question: Where was born the player Leo Barnhorst?** | |
| --- | --- |
| $\alpha$ | **Generation** |
| 0 | Leo Barnhorst was born in **Berlin, Germany**. |
| 100 | Leo Barnhorst was born in **Germany**. |
| 200 | I do not have access to real-time information, including personal details like birthplaces. |
| 300 | I do not have access to real-time information, including personal details like birthplaces. |
| 400 | I couldn't find any information about a player named Leo Barnhorst. |
| 500 | I believe you're asking about**Leo Barnhorst**, a professional soccer player. |
| 600 | I'm unable to provide specific details about the birthplace of a player named Leo Barnhorst. |
| 700 | ?\n\nPlease provide me with the correct spelling of the player's name. |
| 800 | r\n\nI believe you're asking about **Leo Barnhart**, a professional soccer player. |
| 900 | "r\n\nI believe you're asking about **Leo Barnhart**, a professional soccer player. |
| 1000 | r\n\nI believe you're asking about **Leo Barnhart**, a professional soccer player. |
| 1100 | Associate the player Leo Barnhart with the sport of **baseball**. |
| 1200 | criminator: I'm sorry, but I don't have access to real-time information, including personal details like birthplaces. |

Table 6: Gemma 2 2B IT responses to 'Where was born the player Leo Barnhorst?' at different steering coefficient values, $\alpha$ in Equation (4). Leo Barnhorst is unknown for Gemma 2 2B.

# H ACTIVATION PATCHING

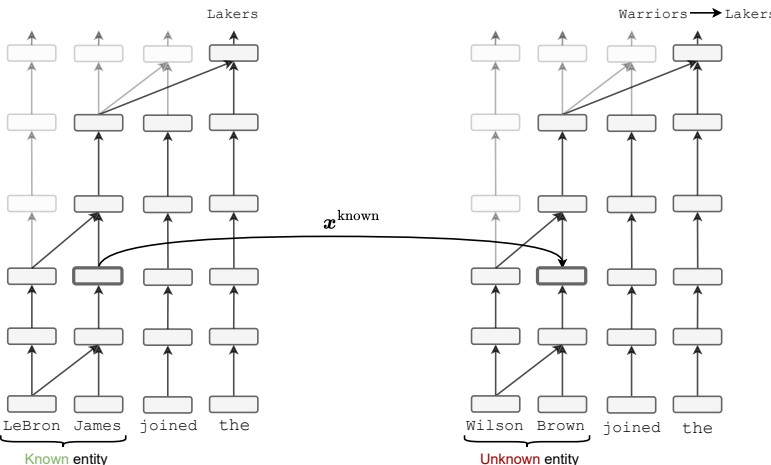

Figure 12: Activation Patching done over the residual stream.

Activation patching (Vig et al., 2020; Meng et al., 2022a; Geiger et al., 2020; Wang et al., 2023) is an intervention procedure performed during a forward pass. We consider a 'clean' input, which in our case is the prompt with a known entity (Figure 12 left). We compute an intermediate hidden state, e.g. the residual steam value at token James, as in Figure 12. Then, we patch this activation at the same site of the forward pass with the corrupted input. In this case, the corrupted input is a prompt with an unknown entity. We can express this intervention using the do-operator (Pearl, 2009) as $f(\mathrm{corr}|\mathrm{do}(\boldsymbol{x}^{\mathrm{unknown}} \leftarrow \boldsymbol{x}^{\mathrm{known}}))$. After the intervention is done, the forward pass continues and the model output is compared with the prediction with the corrupted input. In the experiments in Section 6 we measure the logit difference between the clean (Lakers) and the corrupted predictions (Warriors):

$$\frac{\mathrm{logit}_{\texttt{Lakers}-\texttt{Warriors}}(\mathrm{corr}|\mathrm{do}(\boldsymbol{x}^{\mathrm{unknown}} \leftarrow \boldsymbol{x}^{\mathrm{known}}))}{\mathrm{logit}_{\texttt{Lakers}-\texttt{Warriors}}(\mathrm{clean})} \tag{13}$$

# I ACTIVATION PATCHING ON GEMMA 2 2B

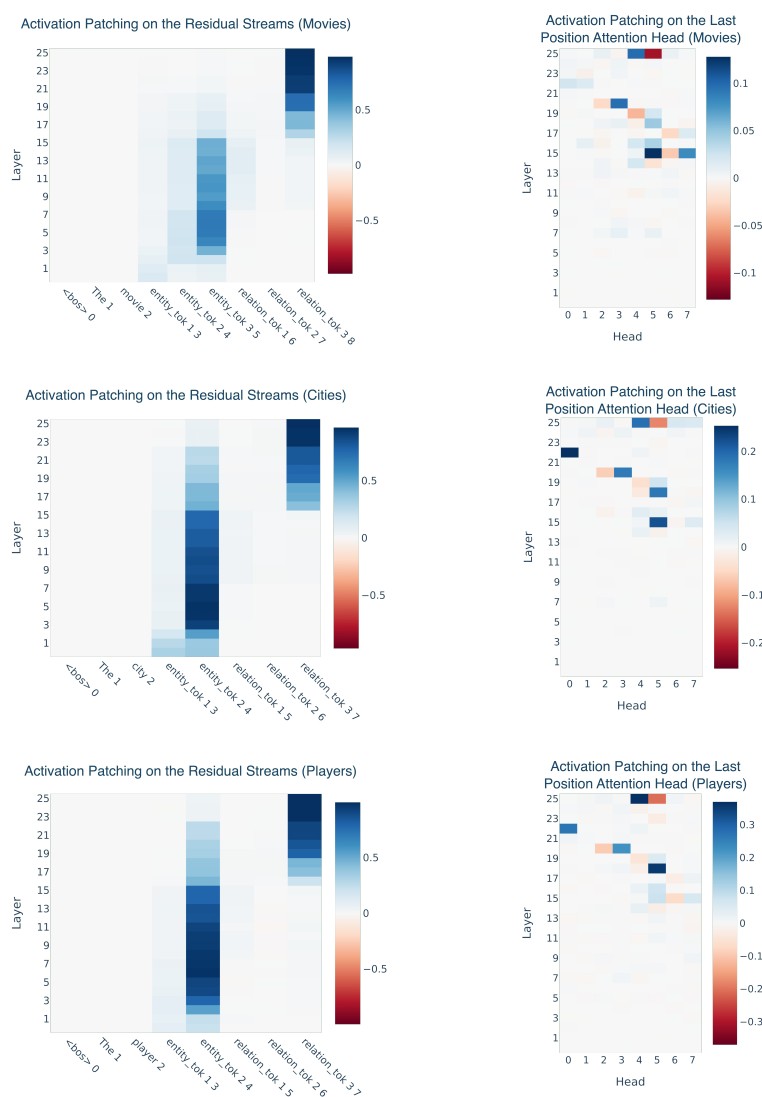

Figure 13: Gemma 2 2B activation patching results on movies (top), players (middle) and cities (bottom).

## J  ACTIVATION PATCHING ON GEMMA 2 9B

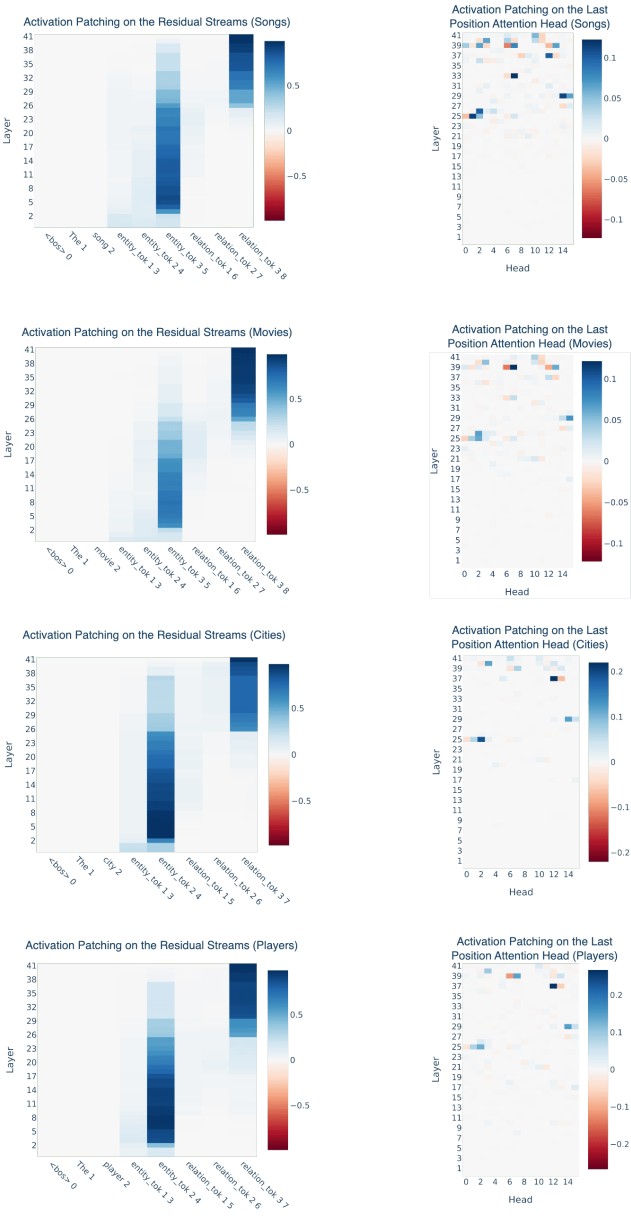

Figure 14: Gemma 2 9B activation patching results on. from top to bottom, song, movies, players and cities.

# K ATTENTION TO LAST ENTITY TOKEN AFTER RANDOM LATENT STEERING

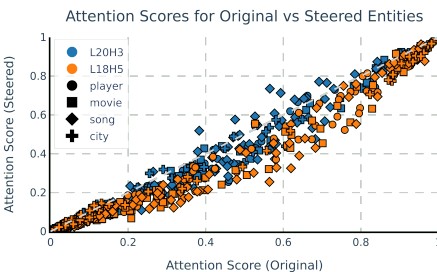

Figure 15: Comparison of attention scores to the last token of the entity after steering with a random SAE latent from Layer 15.

# L ATTRIBUTE EXTRACTION HEADS EXAMPLES

| Head | Entity | Extracted Attributes |
|---|---|---|
| L18H5 | Kawhi Leonard Detmold Boombastic | Clippers, Niagara, Raptors, Westfalen, Lancaster, Volkswagen Jamaican, Reggae, Jamaica, Caribbean |
| L20H3 | Kawhi Leonard Detmold Boombastic | NBA, basketball, Clippers, Basketball Germans, German, Germany, Westfalen reggae, Reggae, Jamaican, music, song |

Table 7: Examples from the top tokens promoted by the attribute extraction heads L18H5 and L20H3 in Gemma 2 2B.

## M    CHANGE OF ATTENTION SCORES TO ENTITIES AFTER STEERING

Gemma 2 2B (Figures 16 and 17), Gemma 2 9B (Figures 18 and 19) and Llama 3.1 8B (Figures 20 and 21) average attention scores to entity tokens after steering with the top known entity latents and top unknown entity latents. Error bars indicate standard deviation. For the known entity latent steering we use prompts with unknown entities, for the unknown entity latent steering we use prompts with known entities. The strength of the steering coefficient is $\alpha = 100$ for Gemma models and $\alpha = 20$ for Llama 3.1 8B. We show heads starting from layer the latent is found, and the steering is done.

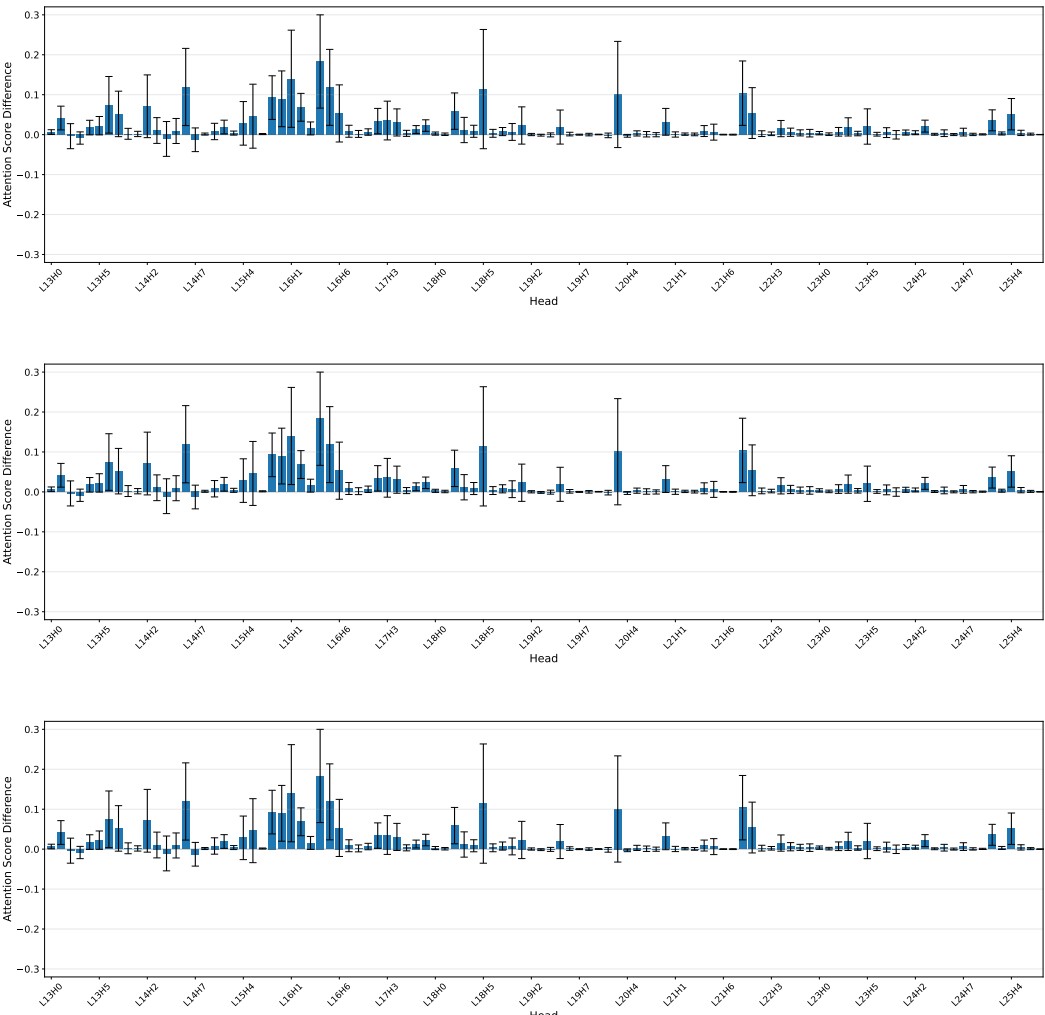

Figure 16: Aggregated attention scores to entity tokens per head in Gemma 2 2B. Steering is done with the top 3 known entity latents (from top to bottom).

We have assessed the statistical significance of attention score changes by comparing steering with entity recognition latents versus with random SAE latents using the same layer and steering coefficient. We conduct t-tests where the null hypothesis states that both steerings would yield identical mean attention scores differences across downstream attention heads. The alternative hypothesis is that known entity latents would increase mean attention scores, while unknown entity latents would decrease them. We tested against 10 different random SAE latents using 100 distinct prompts.

Results indicate that for Gemma 2 2B, Gemma 2 9B and Llama 3.1 8B, steering with the top known entity latent shows statistically significant larger average attention score when compared to random SAE latents on 10/10, 10/10 and 7/10 cases respectively. When steering with the top unknown entity

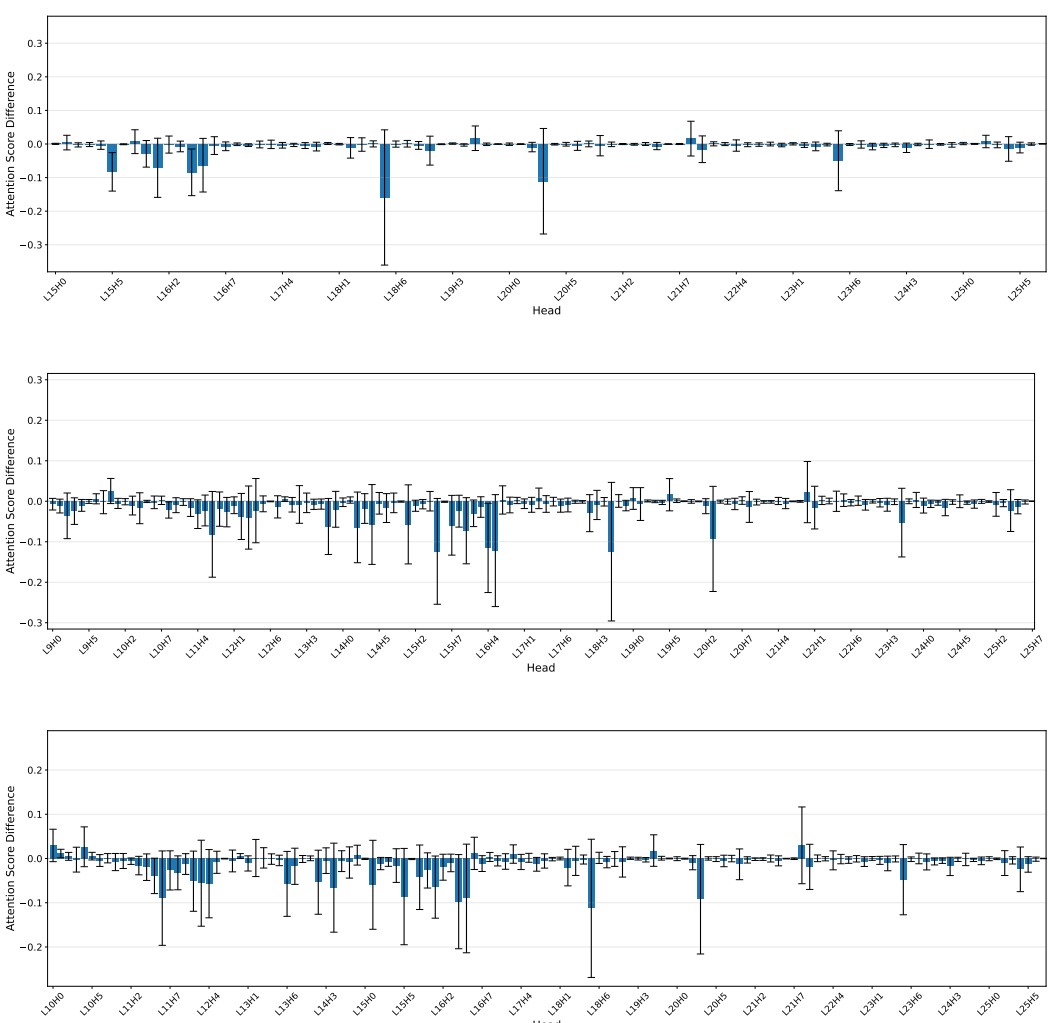

Figure 17: Aggregated attention scores to entity tokens per head in Gemma 2 2B. Steering is done with the top 3 unknown entity latents (from top to bottom).

latent it shows statistically significant lower average attention score when compared to random SAE latents on 9/10, 1/10 and 10/10 cases respectively. As shown in Figure 19 (top), Gemma 2 9B top unknown entity latent doesn't show strong reductions. However the second unknown entity latent shows significant differences in 9/10 tests.

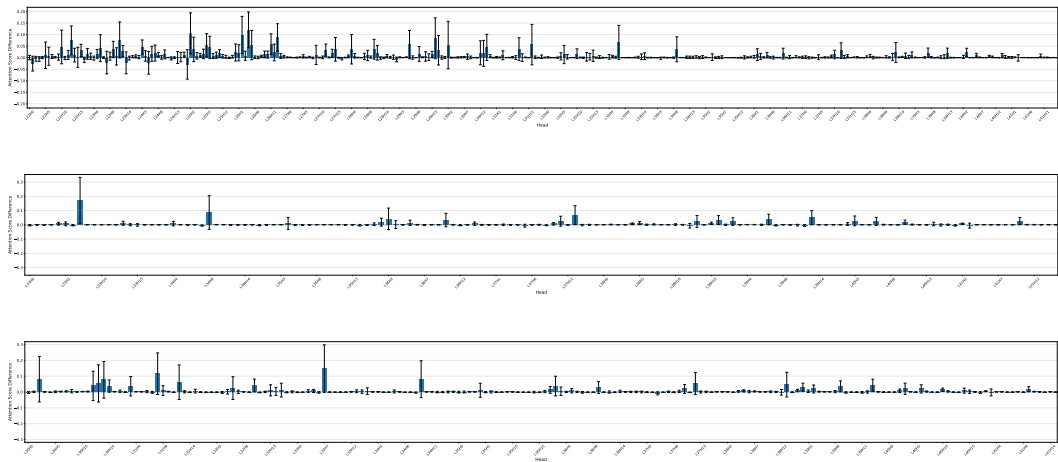

Figure 18: Aggregated attention scores to entity tokens per head in Gemma 2 9B. Steering is done with the top 3 known entity latents (from top to bottom).

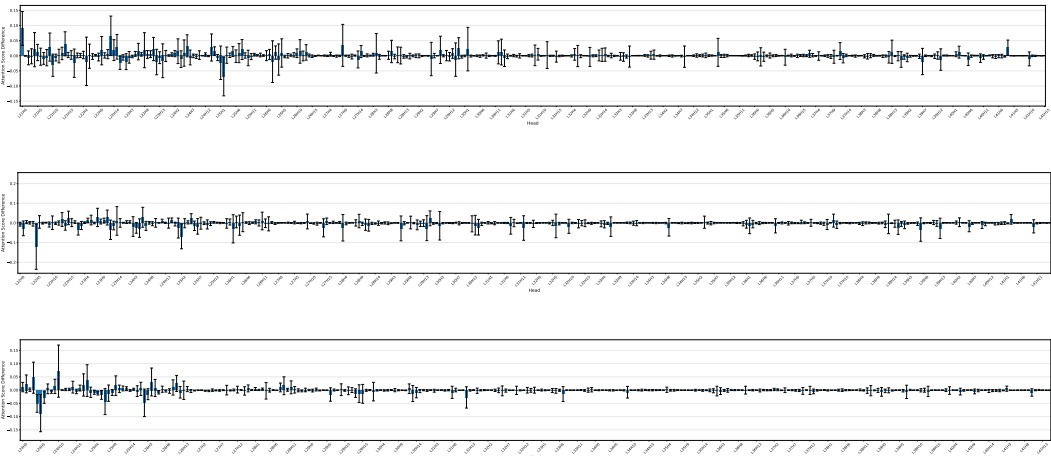

Figure 19: Aggregated attention scores to entity tokens per head in Gemma 2 9B. Steering is done with the top 3 unknown entity latents (from top to bottom).

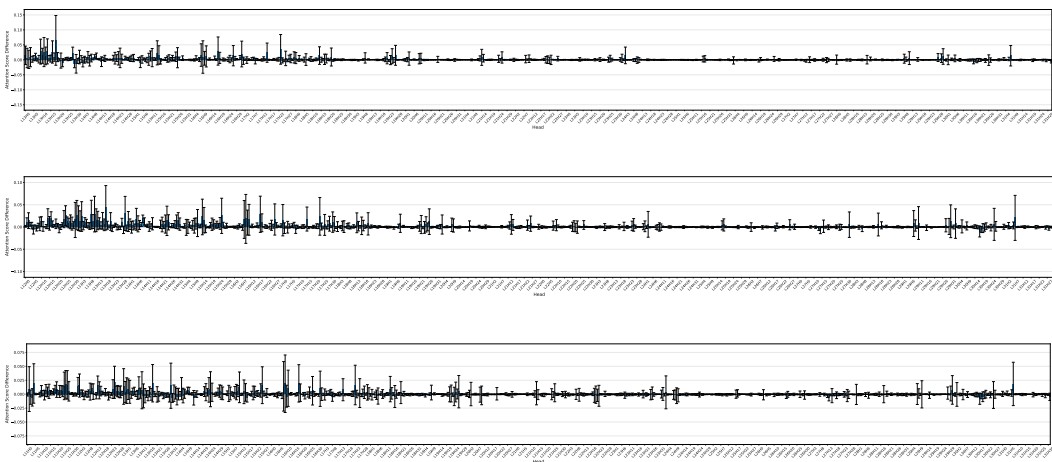

Figure 20: Aggregated attention scores to entity tokens per head in Llama 3.1 8B. Steering is done with the top 3 known entity latents (from top to bottom).

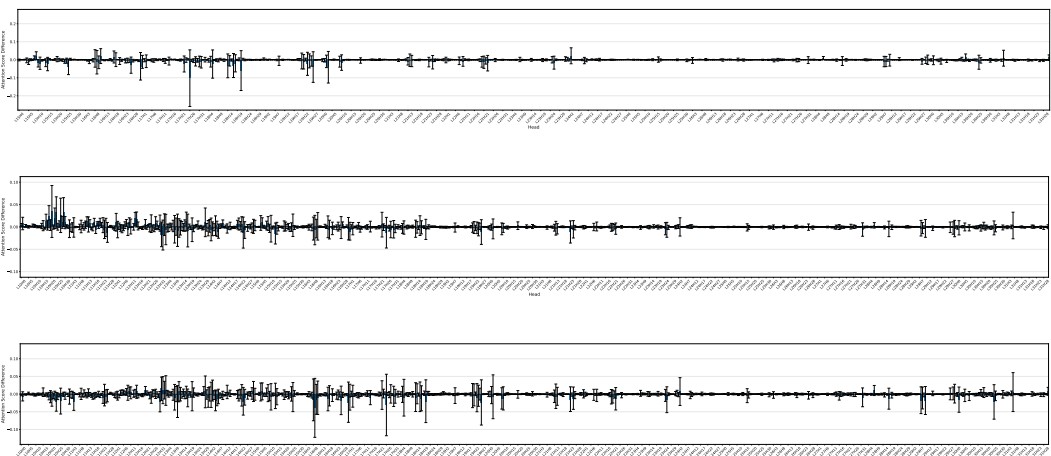

Figure 21: Aggregated attention scores to entity tokens per head in Llama 3.1 8B. Steering is done with the top 3 unknown entity latents (from top to bottom).

## N  GEMMA 2 9B SELF KNOWLEDGE REFLECTION

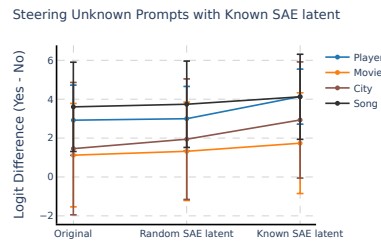
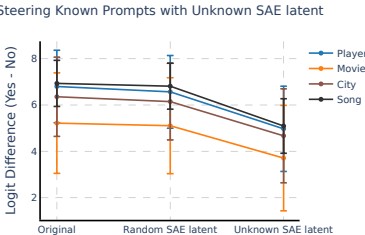

Figure 22: Gemma 2 9B Logit difference between "Yes" and "No" predictions on the question "Are you sure you know the {entity_type} {entity_name}? Answer yes or no." after steering with unknown (left) and known (right) entity recognition latents..

## O  GEMMA 2 9B IT TOP 'UNKNOWN' LATENTS

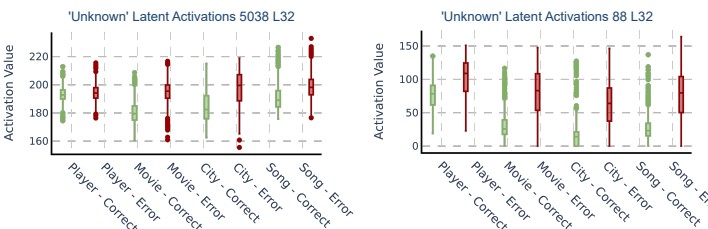

Figure 23: Top 2 Gemma 2 9B IT 'unknown' latents based on the t-statistic score.

## P  GEMMA 2B IT TOP 'UNKNOWN' LATENT WITH SEPARATED ERRORS BASED ON ENTITY TYPE

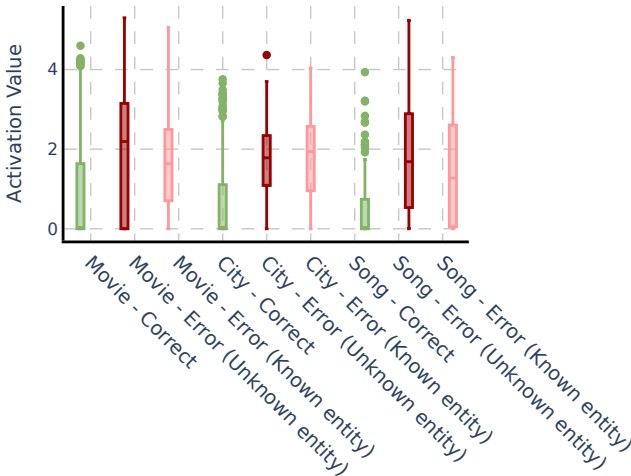

Figure 24: Top 2 Gemma 2B IT 'unknown' latent based on the t-statistic score, with errors divided into known and unknown entities.

## Q   LLAMA 3.1 8B REPLICATION

We extend our experimental analysis to Llama 3.1 8B (Grattafiori et al., 2024), using the SAEs suite from LlamaScope (He et al., 2024), which offers per-layer pretrained SAEs. Following the methodology described in Section 3, we detect both known and unknown entity latents within the model. The distribution of the Top 5 latents across layers (Figure 25) exhibit consistent patterns with previous findings, with the most effective and generalizable latent representations concentrated in the intermediate layers.

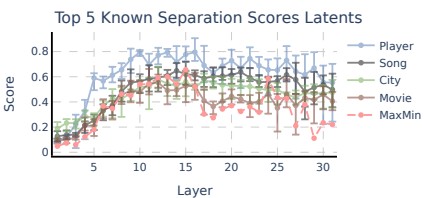 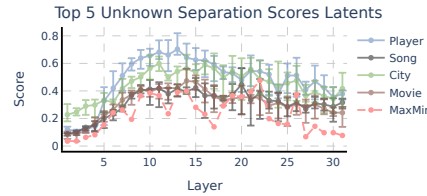

Figure 25: Llama 3.1 8B layerwise evolution of the Top 5 latents, as measured by their known (left) and unknown (right) latent separation scores. Error bars show maximum and minimum scores. MaxMin (red line) refers to the minimum separation score across entities of the best latent. This represents how entity-agnostic is the most general latent per layer. In both cases, middle layers provide the best-performing latents.

Steering experiments using the top unknown entity latent reveal increase refusal rates in the instruction-tuned model (Figure 26). Conversely, when we orthogonalize the model weights with respect to this direction, refusal rates decrease. Since the original model's refusal rate on unknown entity prompts is high (Figure 26 left), we include the refusal rates on prompts with known entities (Figure 26 right). Notably, steering with the top known entity latent did not produce a corresponding decrease in refusals.

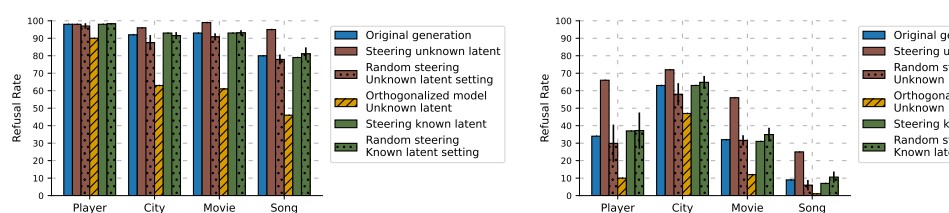

Figure 26: Number of times Llama 3.1 8B refuses to answer in 100 queries about unknown entities (left) and known entities (right). We examine the unmodified original model, the model steered with the known entity latent and unknown entity latent, and the model with the unknown entity latent projected out of its weights (referred to as Orthogonalized model). The mean and standard deviation of steering with 10 random latents are shown for comparison.

Further analysis reveal similar findings to those in Gemma regarding attention patterns: steering with the top known entity latent increases the attention scores to the entity (Figure 27 top), while unknown entity latent steering result in diminished attention scores (Figure 27 bottom).

The replication of our key findings—originally observed in Gemma—across Llama 3.1 8B strengthens our confidence in both our methodological approach and the broader applicability of our results. This generalization is particularly noteworthy given the substantial architectural differences between the two models and their respective SAEs.

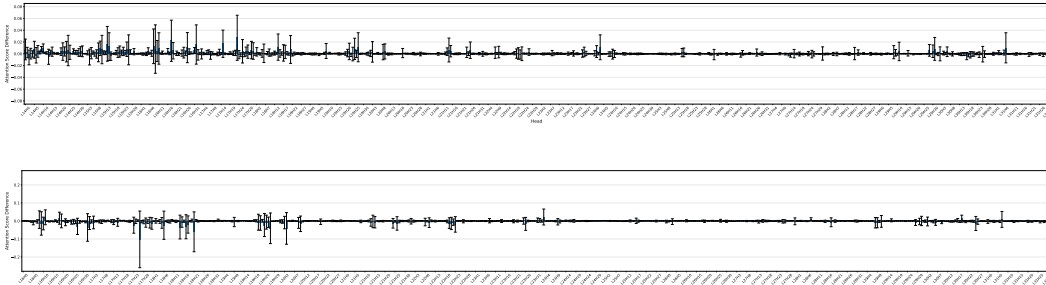

Figure 27: Aggregated attention scores to entity tokens per head in Llama 3.1 8B top known entity (top) and unknown entity (bottom) latents.

## R    ACTIVATION FREQUENCY ON SONGS DATA AFTER KNOWLEDGE CUTOFF

| Model | Known Entity Latent | Unknown Entity Latent |
|-------|---------------------|-----------------------|
| Gemma 2 2B | 6% | 53% |
| Gemma 2 9B | 22% | 55% |
| Llama 3.1 8B | 13.4% | 76% |

Table 8: Activation frequency of each of the top known and unknown entity latents on songs released after knowledge cutoff (August 2024).

## S    TOKEN LIKELIHOOD HYPOTHESIS

An alternative explanation for our observed entity recognition latents is the *token likelihood hypothesis*: the latents might simply encode token likelihood rather than actual knowledge about entities. Under this hypothesis, when processing a token sequence $(t_1, \ldots, t_{i-1}, t_i)$, the activation of our discovered latents at position $i$ could be explained by the model's ability to predict token $t_i$ from previous context. For instance, given a well-known movie title, the model would more easily predict subsequent tokens, potentially triggering what we interpret as 'known entity' latents. Conversely, for unknown entities, lower token likelihood might activate our 'unknown entity' latents. This represents a plausible confounding factor, as tokens comprising well-known entity names are inherently more predictable in the training distribution than those of unknown entities.

To test this hypothesis, we analyze token likelihood on a broad text corpus from the FineWeb dataset (Penedo et al., 2024). For each token position $i$, we compute both the entity recognition latent activations and the probability of the ground-truth token being predicted from position $i - 1$, $p(t_i|t_{<i})$. If the token likelihood hypothesis were true, we would expect strong correlations between these measures.

Our analysis reveals several key findings that challenge this hypothesis:

- Entity recognition latents activate selectively, firing on only a small fraction of tokens (e.g. 0.6% for known entity latents and 0.5% for unknown entity latents in Gemma 2 2B, see Table 9).

- The correlations between latent activations and token prediction probabilities are negligible across all tested models (Figure 28).

- We explored various other potential relationships, including dependencies on the perplexity of surrounding tokens and next-token prediction entropy, finding no substantial correlations.

While we observe that tokens where unknown entity latents activate tend to have lower prediction probabilities compared to the baseline, this effect is modest given the latents' sparse activation patterns. These findings suggest that token predictability alone cannot explain the behavior of our

entity recognition latents, supporting our interpretation that they encode a more sophisticated form of knowledge awareness.

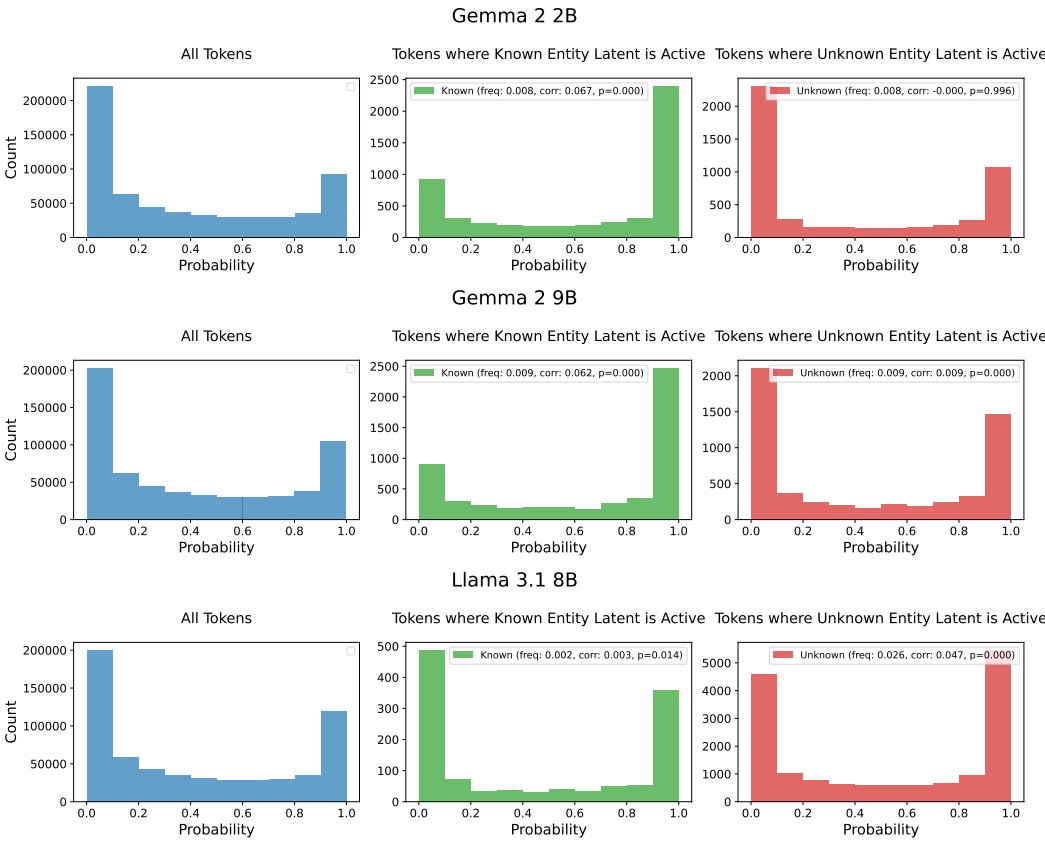

Figure 28: Distribution of ground-truth next-token probabilities for Gemma 2 2B (top), Gemma 2 9B (middle), and Llama 3.1 8B (bottom). For each model, we show three distributions: (left) across all tokens in the dataset, (middle) for tokens where the known entity latent activates, and (right) for tokens where the unknown entity latent activates.

| Model | Latent | Activation Frequency | Correlation with $p(t_i|t_{<i})$ |
|---|---|---|---|
| Gemma 2 2B | Known | 0.006 | 0.067 (p=0.000e+00) |
| | Unknown | 0.005 | -0.000 (p=9.960e-01) |
| Gemma 2 9B | Known | 0.009 | 0.062 (p=0.0e+00) |
| | Unknown | 0.009 | 0.009 (p=1.045e-12) |
| Llama 3.1 8B | Known | 0.002 | 0.003 (p=1.380e-02) |
| | Unknown | 0.026 | 0.047 (p=0.000e+00) |

Table 9: Activation frequency and correlation with conditional next-token probability for top known and unknown entity latents in each model.

