# OpenReview forum: "Do I Know This Entity? Knowledge Awareness and Hallucinations in Language Models"
_ICLR.cc/2025/Conference — ICLR 2025 Oral_

### Official Review · Reviewer_sA3L · 2024-11-03

**Soundness:** 3
**Presentation:** 3
**Contribution:** 3
**Rating:** 8
**Confidence:** 4

**Summary:**

This paper investigates the mechanisms behind LLM hallucinations by using sparse autoencoders -- SAEs to analyze internal model representations. The key finding is the discovery of "entity recognition" directions in the representation space that detect whether a model can recall facts about specific entities. These directions generalize across different entity types (movies, cities, players, songs) and causally influence whether the model refuses to answer questions or hallucinates information. More interestingly, the directions found with a base model can transfer to a chat model and control the model's tendency in refusing to answer questions.

**Strengths:**

- Empirically link the "entity recognition" directions and the models' refusal in answering questions
- The experiment includes clear experimental design using diverse entity types
- Shows interesting discovery that base model mechanisms are repurposed during chat model fine-tuning

**Weaknesses:**

- **Lack of comparison to methods other than SAEs**: If we remove the sparsity of the SAE and reduce the module to a regular AutoEncoder, would the same properties hold? If we further remove the auto encoder and just use a linear probing module, would the same properties hold? Is it really necessary to use sparse and auto encoder modules to get the interpretability? If we borrow the same idea from ITI [1] to steer model towards a more truthful direction in the latent space, can the same property holds? Lack of ablation on the necessity of SAE is a big flaw of this paper.
- **Lack of model diversity**: Only the Gemma models are tested. It would be better if we can see the same findings on LLaMA or other models to show how universal this idea holds.

[1] https://arxiv.org/abs/2306.03341

**Questions:**

- Can you provide more ablation study to show that it's really necessary to use both 1) sparse and 2) auto encoder modules to get the interpretability?

---

> ### Author Response · Authors · 2024-11-25
> **Response to Reviewer sA3L**
>
> Thank you for the constructive review of our paper. We are particularly encouraged by your positive assessment of our experimental methodology and results.
>
> > W1. Lack of comparison to methods other than SAEs
>
> Linear probing can serve as a method to identify meaningful directions within neural representations. However, the strength of SAEs lies in their ability to perform unsupervised discovery, making them a powerful exploratory tool in interpretability research ([Templeton et al., 2024](https://transformer-circuits.pub/2024/scaling-monosemanticity/index.html), [Kissane et al., 2023](https://arxiv.org/abs/2406.17759)). Unlike linear probes, SAEs allow for the identification of novel patterns without requiring a specific training labeled dataset.
>
> Moreover, linear probes may indicate correlations between model representations and the features, but it can’t tell us whether the feature is involved in the model predictions ([Belinkov et al., 2022](https://aclanthology.org/2022.cl-1.7/)). In contrast, SAEs circumvent this issue: the found latents contribute directly to the reconstruction of the representation, providing greater confidence that the identified directions are genuinely used by the model. While SAEs remain a relatively underexplored approach in interpretability research, our work demonstrates their potential as a powerful tool for uncovering unexpected phenomena, as it enabled us to discover the unexpected features presented in this work, offering a valuable case study in their application.
>
> Regarding the sparsity constraint within the SAE, this allows the reconstruction to be a linear combination of a small subset of the learned dictionary features ($W_{\text{dec}}$). It is therefore typically used as a proxy for interpretability. Forcing sparsity is _“based on the intuition that natural data, (such as text or images) may generally be described in terms of a small number of structural primitive”_, we refer to “Why sparseness” section in [Olshause et al., 1997](https://boulderschool.yale.edu/sites/default/files/files/Olshausen_Field_1997.pdf).
>
> > W2. Lack of model diversity
>
> We have run experiments on Llama 3.1 8B, using the SAEs suite from LlamaScope ([He et al., 2024](https://arxiv.org/abs/2410.20526)). Following the methodology detailed in Section 3, we successfully identified both known and unknown entity latents. The layerwise analysis of Top 5 latents follows patterns consistent with our previous observations, with the most effective and generalizable latents emerging in the middle layers. When applying steering with the top unknown entity latent, we also observe increased refusal rates in the instruction-tuned model, as well as a reduction when orthogonalizing the model with that direction. In line with previous results with Gemma, the decrease when steering with the top known entity latent is not clearly observed. Finally, we replicate the observations regarding the disruption of attention to the entity tokens. There is a decrease in attention scores when steering with the unknown entity latent, and an increase when steered with the top known entity latent. All the results can be found in Appendix Q.
>
> The replication of our key findings from Gemma in Llama 3.1 8B gives us increased confidence in the robustness of our methodology and the generalizability of our results. Especially, given the architectural differences between these models, and their associated SAEs.
>
> We thank you for the suggestion, we believe the comparison with other models/SAEs has helped improve our manuscript.
>
>
> > Q1. Can you provide more ablation study to show that it's really necessary to use both 1) sparse and 2) auto encoder modules to get the interpretability?
>
> See response to W1.

---

> ### Author Response · Authors · 2024-12-01
>
> Before the discussion period ends, we wanted to ask the reviewer whether we have addressed your concerns with our work?
>
> We appreciate your feedback, and as a result have added results on LLaMA 3.1 8B and clarified that SAEs played a role in exploratory discovery of the existence of entity recognition, in addition to helping find the entity recognition direction. Exploratory discovery of concepts we didn't expect is a unique advantage of SAEs over probing. We would appreciate clarification on any remaining concerns you have, and ask if you might be open to increasing your score if we have addressed all of them.

---

> > ### Comment · Reviewer_sA3L · 2024-12-02
> >
> > Thanks for the reply! The reply addressed my concerns. I will increase my score to 8.

---

### Official Review · Reviewer_fe4T · 2024-11-03

**Soundness:** 3
**Presentation:** 4
**Contribution:** 4
**Rating:** 10
**Confidence:** 5

**Summary:**

The paper revealed that the knowledge of a large language model can be detected from a sparse latent space. There are several important findings:

- The last token of known and unknown entities have similar latent representation respectively in a sparse latent space across different entity types.

- The latent representation of entities has an influence on LLM generation and steering the representations will affect the refusal and fact recover behaviors of LLMs.

- The latent representation can predict the correctness of generated answers.

**Strengths:**

The findings of the paper (see summary) are well supported by the experiments and can provide valuable scientific insights to the community. The presentation is great and readers can easily follow.

**Weaknesses:**

The experiments are now only for Gemma models. It would be interesting to see the model behavior on other LLMs.

**Questions:**

How does the model size influence the conclusion of the study?

---

> ### Author Response · Authors · 2024-11-26
> **Response to Reviewer fe4T**
>
> We appreciate your positive review and encouragement.
>
> >W1. The experiments are now only for Gemma models. It would be interesting to see the model behavior on other LLMs.
>
> Thank you for this suggestion. We have expanded our analysis to include Llama 3.1 8B using the LlamaScope SAEs release ([He et al., 2024](https://arxiv.org/abs/2410.20526)). The patterns we observed in Llama 3.1 align closely with our findings from the Gemma models, reinforcing the generalizability of our conclusions. The results can be found in Appendix Q.
>
> >Q1. How does the model size influence the conclusion of the study?
>
> We observed particularly pronounced effects in the smaller Gemma 2 model (2B parameters). However, drawing definitive conclusions about the relationship between model scale and our findings requires further investigation.

---

### Official Review · Reviewer_kSkD · 2024-11-03

**Soundness:** 4
**Presentation:** 4
**Contribution:** 3
**Rating:** 8
**Confidence:** 4

**Summary:**

Using sparse autoencoders (SAEs), this paper shows the existence of directions in the latent space of LLMs responsible for entity recognition i.e. detecting whether the entity is a known entity or not. The paper shows that these identified latents are causally responsible for LLMs refusing to answer questions. The paper shows how these latent directions are responsible for regulating attention scores, and also responsible for directly expressing knowledge uncertainty. All experiments are conducted with Gemma 2 models (2B and 9B) where SAEs are available.

**Strengths:**

1. The result showing that directions identified using base model, causally affect refusal in chat-finetuned models is very interesting — the transfer from base to instruction/chat finetuned model is not obvious, and indicates a mechanism for how refusal finetuning can generalize beyond the specific entities it was finetuned on.

2. All of the experiment and results figures seem thorough and convincing (e.g. error bars, useful and clear visualizations etc.)

3. The paper also shows that interestingly, the same entity recognition directions regulate the model's attention mechanism (the factual recall mechanism) and directly express knowledge uncertainty (line 355, although the effect is not as strong).

**Weaknesses:**

1. Definition of known and unknown entities — the paper categorizes entities as known if it gets at least two attributes right, and unknown if it gets all attributes wrong. While this is fine, it does leave open the possibility that what is currently categorized as unknown is actually known by the model and can perhaps be elicited with a better prompt or few-shot examples. Alternatives to the current categorization could be – (a) checking across a diverse set of prompts to decide known and unknown as in https://arxiv.org/abs/2405.05904 ; (b) relying on n-gram matching in the pretraining data (would need to use LLMs with open pretraining data) ; (c) focusing on entities which occur after the knowledge cutoff for the unknown category.

2. Explanation for existence of generalized latents that distinguish between known and unknown entities — the paper currently suggests that this is evidence of ‘self-knowledge’. An alternative and much simpler explanation which currently can’t be ruled out is that these latents are activated on high likelihood tokens (known) and low likelihood tokens (unknown). It would be useful to show some kind of control experiment to show that is not the case.

3. (minor) In general, I also find it hard to directly interpret the results in Fig 2. The min-max value peaks at around 0.4 for unknown. How high should that value be to conclude the existence of generalized latents? I feel like if the set of the entities is small enough and the set of latents large enough, then there might always exist such latents? (Fwiw the additional results later do make it convincing that these are generalized latents)

**Questions:**

1. Formatting — the bottom margin seems much bigger than in the ICLR template?

2. Line 160, on average how many attributes does each entity have?

3. I also couldn’t find details about how many entities of each type was used in the analysis.

4. Figure 5 — Prior work has shown that LLMs are better calibrated when using few-shot examples (e.g. https://arxiv.org/abs/2207.05221) — have you tried changing the prompt in 10 (line 359) to see if the intervention has a bigger effect in that case?

5. Line 35 Relevant citation - https://arxiv.org/abs/2310.15910

6. Sec 7 — prior work (e.g. https://arxiv.org/abs/2310.18168) also shows that linear probes can be trained to predict truthfulness / correctness before the answer (just from the question), which would indicate similar findings as in section 7.

---

> ### Author Response · Authors · 2024-11-26
> **Response to Reviewer kSkD**
>
> We thank you for your thorough review. We appreciate the positive comments, as well as the thoughtful points raised.
>
> > W1. Definition of known and unknown entities. It does leave open the possibility that what is currently categorized as unknown is actually known by the model and can perhaps be elicited with a better prompt or few-shot examples.
>
> The primary objective is not to create perfect distinctions between known and unknown categories. Rather, we aim to verify that these latents can distinguish between the two types of entities. We believe minor classification errors are acceptable, as they don't impact the core conclusions of the paper. We have now raised this point in the paper.
> As you suggested, to validate the robustness of the entity recognition latents we identified, we conducted additional testing using a list of 283 songs released after August 2024 (post-model training). The frequency of activation of the entity recognition latents across models are:
>
> | Model | Known Entity Latents | Unknown Entity Latents |
> |:--------:|:-------------------:|:---------------------:|
> | Gemma 2 2B | 6% | 53% |
> | Gemma 2 9B | 22% | 55% |
> | Llama 3.1 8B | 13.4% | 76% |
>
> We acknowledge that this validation set may contain repeated song titles from previous years that overlap with pre-training data. However, the consistent activation patterns across models support the robustness of our identified latents in distinguishing knowledge states.
>
> > W2. Simpler explanation which currently can’t be ruled out is that these latents are activated on high likelihood tokens (known) and low likelihood tokens (unknown).
>
> Thank you for this suggestion. We agree that token likelihood represents a compelling potential confounder, as known entities are inherently more predictable than unknown ones. To address this concern, we have conducted an extensive investigation detailed in Appendix S ("Token Likelihood Hypothesis").
>
> The entity recognition latents discussed in the paper were selected after filtering out those that activate on more than 2% of random tokens from the Pile. To further evaluate the Token Likelihood Hypothesis, we analyzed these latents' activations on a sample of the FineWeb dataset. Our analysis reveals that these latents are highly sparse, activating on only 0.1-2.6% of tokens, and show minimal correlations with token prediction probabilities (Table 9).
>
> We have also investigated more complex dependencies by examining relationships with prediction entropy and perplexity of the surrounding tokens, finding no substantial correlations. We believe that these results provide sufficient evidence to conclude that the latents we find encode a more sophisticated form of entity recognition beyond simple token predictability.
>
> > W3. Hard to directly interpret the results in Fig 2. How high should that (min-max) value be to conclude the existence of generalized latents?
>
> Unfortunately, there is no specific min-max value which we can take as reference, especially since the number of labeling errors are unknown. However, a particularly noteworthy finding was the systematic progression of scores across layers in all models, which suggests non-random, structured formation of entity recognition features. Our study includes approximately 35,000 distinct entities across four types. However, the entity recognition capabilities we observed extend well beyond these core categories. As demonstrated in Tables 4 and 5 (Appendix B), the identified latent representations largely activate across a diverse range of entities, including corporations, scientific figures, artists, and cultural institutions, suggesting high generalizability.
>
>
> > Q2. On average how many attributes does each entity have?
>
> On average each entity type has 4.75 attributes. We refer to Table 3, Appendix A.
>
> > Q3. I also couldn’t find details about how many entities of each type was used in the analysis.
>
> | Entity Type | Number of entities | Attributes |
> |------------|-------------------|------------|
> | Player | 7,487 | Birthplace, birthdate, teams played |
> | Movie | 10,895 | Director, screenwriter, release date, genre, duration, cast |
> | City | 7,904 | Country, population, elevation, coordinates |
> | Song | 8,448 | Artist, album, publication year, genre |
>
> This information is available in Table 3, Appendix A.
>
> > Q4. Have you tried changing the prompt in 10 (line 359) to see if the intervention has a bigger effect in that case?
>
> Thanks for the recommendation! As suggested in the paper, we will add few-shot examples and check if the effect is more pronounced.
>
> > Q6 and Q7.
>
> Thanks for the suggestions, we have included them in the paper.

---

> > ### Comment · Reviewer_kSkD · 2024-11-26
> >
> > Thanks for a detailed response! I think my main concerns regarding how unknown entities are identified, and that likelihood might explain the results, have been addressed. Given that, I'm happy to increase the score.

---

> > > ### Author Response · Authors · 2024-12-01
> > > **Thank you!**
> > >
> > > We are pleased that our responses adequately addressed your concerns. Thank you for the improved score.

---

### Official Review · Reviewer_pFUP · 2024-11-05

**Soundness:** 4
**Presentation:** 4
**Contribution:** 3
**Rating:** 10
**Confidence:** 4

**Summary:**

This paper discusses some interesting features found in sparse features extracted from Gemma 2 2B and 9B models with the Gemma Scope SAEs. These features, when queried on the final token of an entity mention, are predictive of whether the model correctly recalls facts about that entity in a synthetic task constructed from Wikidata. Some features generalize across multiple (four) entity types, and interventional experiments on chat-tuned models find that they mediate refusals in these models, suggesting that RL training may preferentially lead models to repurpose these mechanisms developed in pretraining. Further analysis shows that these features, when activated, disrupt the normal mechanisms for factual recall by blocking attribute extraction heads. A final experiments identifies SAE latents in a similar spirit which are predictive of model mistakes.

**Strengths:**

* The paper is excellently written. The prose is very clear and the arguments are well structured.
* The paper fruitfully pursues multiple lines of evidence for its core claims about the latents it identifies in the model it studies. Having correlational and causal analysis on multiple fronts including with the RLHF'd model makes the central idea feel pretty solid.
* The paper is well scoped.

**Weaknesses:**

Some of the claims of the paper feel not quite adequately supported by the experiments. I'm not really doubtful of the reasonableness of the claims but I think the paper should quantify and qualify them better.
* I think the choice to describe these latents as representing "self-knowledge" is a little bit dicey. The description only clearly applies to the chat model. For the base model I think a sufficient explanation of the evidence with fewer assumptions is to say something like they're latents for "entities it knows about" versus "entities it doesn't know about or treats as novel". You could make progress disentangling this from 'self-knowledge' by looking at how these latents behave on fictional narrative versus newswire versus encyclopedic text (the former two should have been written after training cutoff). I would expect the 'known entity' latents to fire increasingly more as you move from fiction->news->encyclopedia and opposite for 'unknown entity' latents. Analyzing these results could help us understand why the features are learned, e.g., if the the model needs to know whether it is writing stories or stating facts.
* The paper claims that the features generalize over multiple entity types. This is a binary statement but of course is about a graded judgment regarding the amount of generalization. But there is no quantitative claim in the paper about the degree to which this is true in an absolute sense—only that it is _more_ true in the middle layers. That doesn't mean that it's particularly true anywhere. The maxmin metric makes sense for relative comparisons but is difficult to understand on an absolute level. Indeed, when you select the "most general" latents based on this metric it's unclear if these features are actually general across more than the four entity types you use or if you may just be selecting on noise. To measure this, you should measure what happens when you select latents based on their generality across a subset of entity types, and test their separation score on held-out entity types. Some measure of how predictive this selection is of separation for new entity types gives an actual measure of the generality of the latent.
* In the mechanistic analysis (p.6–7) the paper points to figures which seem to show that attention to an entity is decreased when the unknown entity latents are steered (increased). It's not clear to me from the text if this is averaging across all attention scores or some subset ("these attention heads", L345, reads to me like the latter, but I'm not sure which subset) and the results of this experiment aren't quantified. There should be some straightforward measure here which you can use to quantify the suppression effect here (which will also allow you to claim statistical significance — you do say "significantly" on L346 without a relevant quantity or p-value, which leaves it unclear if you actually mean statistical signifiance)
* The results in Figure 5 seem dubious. They are really hard to read because of the overlap of all of the confidence intervals but overall nothing significant seems to be happening here. I think you'd need to run this study on much more data to get something interesting/clearly meaningful and I also find this the least interesting or informative experiment in the paper. Mild recommendation to appendicize it. (and strong rec to make the statistical claims precise either way: is the effect you're talking about significant or not?)

**Questions:**

* The main text doesn't say which direction d_j you orthogonalize out on p.6. (caption says it but main text description of what you do is disjointed.)
* Equation 11's use of the word `model` is weird. I assume it's supposed to be a metavariable, but I don't feel like the red highlight conveys this (since it's used for something else above). You already frame it as an example—why not use the actual example correct or incorrect answer there? I think it'd be fine to have a line break in there if the problem is making it fit on the page.
* Do you have a succinct/clear and empirically grounded explanation of why you use (your formulation of) latent separation scores for the first analysis and t-statistics for the second? It sounds like the features often activated on both categories and the separation scores probably just weren't as good in the latter case, is that right? Seems worth being explicit about in the paper.
* Can you clarify in the paper how you use the train/test split in the analysis in Section 7? I assume the graphs in Figure 6 plot results for the test set. Also again please give statistics on your results and discuss significance.
* Do you think you can add some discussion in the paper about how your approach can be generalized to other kinds of features? The targeted investigation you do is interesting on its own but I think it'd be great to highlight if there's anything else that someone can expect to discover by adapting your analysis setup to different datasets.

[Edit: many of the weaknesses and questions have been answered adequately by the reviewers, so I've upped the soundness rating from 3 to 4 and overall rating from 8 to 10.]

---

> ### Author Response · Authors · 2024-11-27
> **Response to Reviewer pFUP (1/2)**
>
> We appreciate your careful review and validation of our experimental approach and findings.
>
> > W1.  I think the choice to describe these latents as representing "self-knowledge" is a little bit dicey.
>
> We refer to _knowledge awareness_ as ‘_detecting whether the model can recall facts about the entity_’ (lines 80-81). Our observations suggest this awareness mechanism causally affects the refusal behavior of the chat models. However, we emphasize that our definition of knowledge awareness focuses specifically on the detection process, not the subsequent response behavior. We approach the term "self-knowledge" with appropriate caution and, in line with your feedback, we have tempered our claims regarding “self-knowledge” in the revised version of the paper.
>
> > W2. The maxmin metric makes sense for relative comparisons but is difficult to understand on an absolute level. Indeed, when you select the "most general" latents based on this metric it's unclear if these features are actually general across more than the four entity types you use.
>
> While we did the quantitative analysis on four types of entities, we observe the generality of entity recognition latents extend beyond those. Looking at examples of sentences generated by Claude, included in Tables 4 and 5 (Appendix B), we see how these latents pick up on all sorts of entities—everything from companies and scientists to artists and museums.
>
> > W3. In the mechanistic analysis… It's not clear to me from the text if this is averaging across all attention scores or some subset.
>
> We apologize for the confusion, we refer to the aggregated attention scores to the entity tokens of all heads downstream from the latent. We have clarified this in the revised version of the manuscript and shown the differences in attention scores for all downstream heads in Appendix M.
>
> >There should be some straightforward measure here which you can use to quantify the suppression effect here.
>
> Since the magnitude of change varies with the steering coefficient, we have assessed the statistical significance of attention score changes by comparing steering with entity recognition latents versus with random SAE latents using the same layer and steering coefficient. We conduct t-tests where the null hypothesis states that both steerings would yield identical mean attention scores differences across downstream attention heads. The alternative hypothesis is that known entity latents would increase mean attention scores, while unknown entity latents would decrease them. We tested against 10 different random SAE latents using 100 distinct prompts.
>
> Results indicate that for Gemma 2 2B,  Gemma 2 9B and Llama 3.1 8B, steering with the top known entity latent shows statistically significant larger average attention score when compared to random SAE latents on 10/10, 10/10 and 7/10 cases respectively. When steering with the top unknown entity latent it shows statistically significant lower average attention score when compared to random SAE latents on 9/10, 1/10 and 10/10 cases respectively. As shown in Figure 19, Gemma 2 9B top unknown entity latent doesn’t show strong reductions. However, the second unknown entity latent shows significant differences in 9/10 tests.
>
> These results demonstrate consistent patterns in how steering affects attention scores across different model architectures, with particularly strong effects observed in Gemma 2 2B and Llama 3.1 8B.
>
> > W4. The results in Figure 5 seem dubious. They are really hard to read because of the overlap of all of the confidence intervals
>
> The results in Figure 5 show a moderate but consistent correlation between the effect of steering with entity recognition latents and model self-knowledge expressions across all entities. While the effect size is modest, the directional movement in yes/no motivates further investigation. As you suggest, we plan to validate these findings through significance testing against random baselines using an expanded dataset.
>
> > Q1. The main text doesn't say which direction d_j you orthogonalize out on p.6.
>
> We orthogonalize the model with the top unknown entity latent. We have now made it more clear it in the paper.
>
> > Q2. Equation 11's use of the word model is weird.
>
> We refer to the token ‘model’, which is one of the end-of-instruction tokens in Gemma chat models. The example (11) is the exact prompt presented to Gemma. We have substituted the red highlighting to avoid confusion.

---

> ### Author Response · Authors · 2024-11-27
> **Response to Reviewer pFUP (2/2)**
>
> > Q3. Do you have a succinct/clear and empirically grounded explanation of why you use (your formulation of) latent separation scores for the first analysis and t-statistics for the second?
>
> Our initial analysis on entities examines a binary distinction between known and unknown entities. This is motivated by its applicability to hallucinations, as an ideal model should refuse to answer if presented with an unknown entity. Later, in Section 7, we explore a more nuanced aspect: the model's degree of uncertainty. Rather than treating uncertainty as a binary state, we conceptualize it as existing along a continuous spectrum. For instance, the model might still express some uncertainty even when answering correctly and therefore, we believe measuring the difference in activation means through the t-static seems more appropriate.
>
>
> > Q4. Can you clarify in the paper how you use the train/test split in the analysis in Section 7?
>
> We divide the dataset of prompts into train/validation/test sets (50%, 10%, 40%). We use the training set to compute t-statistics and select the top latent. We use the test set to extract the left panel of Figure 6 and the AUROC and F1 scores. The validation set is used to choose the optimal decision threshold for computing the F1 score. We have clarified it in the paper.
>
> > Q5. Do you think you can add some discussion in the paper about how your approach can be generalized to other kinds of features?
>
> We appreciate your suggestion. Our methodology for feature detection from SAEs can indeed be generalizable across different feature types. The framework we present in Section 3 enables the identification of binary features, while the use of the t-statistic score (in Section 7) instead of the latent separation scores extends this capability to features that can be considered continuous, such as model uncertainty. We have incorporated these clarifications in the revised manuscript to better emphasize the broad applicability of our approach.

---

> ### Comment · Reviewer_pFUP · 2024-11-30
> **Sounds great!**
>
> This all sounds great. Thanks for your clarifications and improvements to the manuscript. Upping my Soundness rating to 4 and overall rating to 10. The overall score does feel a bit extreme as I would normally reserve that for groundbreaking work and the contribution here is a bit more focused. But the work seems super solid to me and I think is worth considering at least for a spotlight.

---

> > ### Author Response · Authors · 2024-12-01
> > **Thank you!**
> >
> > We thank you again for your thoughtful review and consideration of our work. We are grateful for both the increased scores and the spotlight recommendation.

---

### Meta-Review · Area_Chair_SL2Q · 2024-12-24

**Metareview:**

This is a very solid work. It begins with the key observation that hallucination in large language models (LLMs) is closely linked to their ability (or inability) to recall knowledge about entities. It then introduces a novel approach, leveraging sparse autoencoders to identify latent directions within the LLMs' representation space that are responsible for entity recognition. Comprehensive experiments demonstrate that these directions causally influence the LLMs' refusal to answer questions. Furthermore, the study investigates how these latent directions regulate attention scores, and predict knowledge uncertainty and incorrect answers.

That said, there are a few aspects raised that make the work not yet perfect:
- The predictive accuracy regarding incorrect answers is not thoroughly explored. In studies within this field, it is essential to demonstrate direct applications, such as automatic hallucination detection or revision, using public datasets and to compare these results with other existing approaches.
- The scope of the study is somewhat narrow, focusing exclusively on knowledge about entities. Also the proposed approach does not address scenarios of hallucinations on known facts.
- The number of LLMs evaluated in the study is relatively limited, although the authors later provide evidence supporting the generalizability of their approach.

**Additional Comments On Reviewer Discussion:**

No new points raised during rebuttal.

---

### Decision · Program_Chairs · 2025-01-22

Accept (Oral)